# A quantitative mass spectrometry-based approach to monitor the dynamics of endogenous chromatin-associated protein complexes

Evangelia K. Papachristou [1], Kamal Kishore[1], Andrew N. Holding[1], Kate Harvey[2], Theodoros I. Roumeliotis[3], Chandra Sekhar Reddy Chilamakuri[1], Soleilmane Omarjee[1], Kee Ming Chia[2], Alex Swarbrick[2,4], Elgene Lim[2,4], Florian Markowetz [1], Matthew Eldridge[1], Rasmus Siersbaek[1], Clive S. D'Santos[1] & Jason S. Carroll[1]

Understanding the dynamics of endogenous protein–protein interactions in complex networks is pivotal in deciphering disease mechanisms. To enable the in-depth analysis of protein interactions in chromatin-associated protein complexes, we have previously developed a method termed RIME (Rapid Immunoprecipitation Mass spectrometry of Endogenous proteins). Here, we present a quantitative multiplexed method (qPLEX-RIME), which integrates RIME with isobaric labelling and tribrid mass spectrometry for the study of protein interactome dynamics in a quantitative fashion with increased sensitivity. Using the qPLEX-RIME method, we delineate the temporal changes of the Estrogen Receptor alpha (ERα) interactome in breast cancer cells treated with 4-hydroxytamoxifen. Furthermore, we identify endogenous ERα-associated proteins in human Patient-Derived Xenograft tumours and in primary human breast cancer clinical tissue. Our results demonstrate that the combination of RIME with isobaric labelling offers a powerful tool for the in-depth and quantitative characterisation of protein interactome dynamics, which is applicable to clinical samples.

[1] Cancer Research UK Cambridge Institute, University of Cambridge, Robinson Way, Cambridge CB2 0RE, UK. [2] Garvan Institute of Medical Research, Darlinghurst, Sydney NSW 2010, Australia. [3] Wellcome Trust Sanger Institute, Wellcome Genome Campus, Cambridge CB10 1SA, UK. [4] St Vincent's Clinical School, UNSW, Sydney NSW 2052, Australia. These authors contributed equally: Kamal Kishore, Andrew N. Holding. Correspondence and requests for materials should be addressed to R.S. (email: Rasmus.Siersbaek@cruk.cam.ac.uk) or to C.S.D'S. (email: Clive.D'Santos@cruk.cam.ac.uk) or to J.S.C. (email: Jason.Carroll@cruk.cam.ac.uk)

Deciphering the role and the organisation of dynamically regulated protein networks is critical for the accurate molecular characterisation of biological systems[1]. Over the last decade, the advancements made in mass spectrometry-based proteomics have enabled the rapid analysis of complex protein samples obtained from co-immunoprecipitation assays, providing a powerful tool for the study of protein interactions and protein complexes[2]. In this regard, the first systematic efforts to generate human protein interactome maps using yeast two-hybrid[3–5] have been recently complemented by studies utilising large-scale Affinity Purification followed by Mass Spectrometry analysis (AP-MS)[6,7]. Additionally, the integration of AP-MS with quantitative approaches has enabled the study of stoichiometric changes in protein complexes[8]. More recently, the use of chemical crosslinking combined with mass spectrometry has provided information about endogenous protein assemblies in a proteome-wide scale[9].

Gene regulation relies on the coordinated action of transcription factors and co-regulator complexes that control transcriptional activation at promoters or enhancers. To gain insight into the complex interactions between such regulators, the combination of Chromatin Immunoprecipitation (ChIP) with mass spectrometry has been used to study the composition of chromatin-associated complexes[10–12]. In line with this strategy we have previously developed RIME (Rapid Immunoprecipitation Mass spectrometry of Endogenous proteins)[13], a method which has several advantages for the analysis of protein interactomes[14]. RIME provides a sensitive and rapid approach for the identification of protein complexes from low amounts of starting material and importantly involves purification of endogenous protein, rather than the use of exogenous tagged approaches.

In the present study, we have established a modified RIME assay to monitor the dynamics of chromatin-associated complexes using a quantitative multiplexed workflow (quantitative Multiplexed Rapid Immunoprecipitation Mass spectrometry of Endogenous proteins or qPLEX-RIME). Specifically, we combine RIME with isobaric labelling using Tandem Mass Tags (TMT-10plex)[15,16], peptide fractionation and MultiNotch MS3 analysis[17]. This combination allows the simultaneous analysis of multiple conditions and biological replicates with high sensitivity in a single experiment. Additionally, we have developed a data analysis workflow termed quantitative Multiplexed analyzer (qPLEXanalyzer) that permits statistical analysis of the quantitative interactome data and the identification of differential interactions.

As a proof-of-concept, we apply the qPLEX-RIME method to discover the temporal changes of Estrogen Receptor alpha (ERα) interactors in breast cancer cells treated with 4-hydroxytamoxifen (OHT) and to identify the ERα interactome in human patient-derived xenograft (PDX) tumours and in human breast cancer tissues. Our data demonstrate that the qPLEX-RIME method combines multiplexity with quantitative accuracy and increased sensitivity, to enable the in-depth characterisation of dynamic changes in chromatin-associated protein complexes in vitro and in vivo.

## Results

**The qPLEX-RIME workflow**. The qPLEX-RIME approach combines the RIME method[13,14] with multiplex TMT chemical isobaric labelling[15,16] to study the dynamics of chromatin-associated protein complexes. The workflow starts with a two-step fixation procedure using disuccinimidyl glutarate (DSG) and formaldehyde (FA) that has been previously applied in combination with ChIP assays to capture transient interactions more efficiently[12,18]. A specific antibody against the target protein is used for immunoprecipitation, followed by proteolysis, TMT-10plex peptide labelling and fractionation. The main steps of the qPLEX-RIME method are shown in Fig. 1. The main utility of the qPLEX-RIME method is the quantification of changes in the composition of protein complexes in response to cell perturbation and/or in variable genomic backgrounds (e.g. different cell lines or mutated conditions) using multiple biological replicates in a single experiment. Also proteins that are significantly and specifically associated with the bait protein can be discovered in the same analysis using appropriate negative controls, such as IgG pull-downs. For the downstream data analysis, we have developed a comprehensive bioinformatics workflow (qPLEXanalyzer) that includes data processing, visualisation, normalisation and differential statistics . In addition to the qPLEXanalyzer R package, the complete qPLEX-RIME and full proteome data sets of this work are included in the qPLEXdata R package. Both packages can be found at (https://doi.org/10.5281/zenodo.1237825) and a detailed description of the pipeline and the applications is provided in Supplementary Notes 1 and 2.

**Characterisation of the ERα interactome in MCF7 cells**. We first applied qPLEX-RIME to assess whether we could successfully identify the ERα interactome in asynchronous MCF7 breast cancer cells. To this end, we performed ERα qPLEX-RIME pull-downs in five independent biological replicates. An equal number of matched IgG control samples were prepared. In this experiment we used single crosslinking with FA, to permit a comparison with previously published approaches[13]. In addition to the qPLEX-RIME, we included a standard non-quantitative ERα RIME experiment with matched IgG controls (Supplementary Data 1).

The qPLEX-RIME raw data processing quantified 2955 proteins across the multiplexed set of all positive and negative samples at peptide false discovery rate (FDR) <1% (Supplementary Data 2). To test the efficiency of the method in capturing and quantifying previously described ERα-associated proteins, we compiled a list of known ERα interactors from BioGRID[19] and STRING[20] resources. For BioGRID, we used only a subset of 386 proteins identified by high-throughput assays that are similar to the approach used here and for STRING we used only experimental associations (383 proteins, score > 200). Noteworthy, only 37 proteins were common between the two reference subsets. The qPLEX-RIME method identified 295 (76%) and 171 (45%) of the known ERα-associated proteins from BioGRID and STRING, respectively, of which 225 (58%) and 154 (40%) showed positive enrichment at adj. $p$-value < 0.1 (Limma moderated $t$-test) (Fig. 2a). Specifically, we found known co-regulators (e.g. EP300, NCOA3, CBP, NRIP1, TRIM24, GREB1, RARα, NCOR2 and HDACs[13,21–25]), ERα-associated pioneer factors (e.g. FOXA1[26] and AP-2γ[27]), and putative pioneer factors (e.g. GATA-3[28]) with significant enrichment in the ERα samples (Fig. 2b).

ERα was one of the most significantly enriched proteins identified with 19 unique peptides (Fig. 2c), which is consistent with previously published ERα RIME experiments[14]. A comparison between the non-quantitative RIME and the qPLEX-RIME data showed that 302 of the 323 (93%) proteins identified as ERα-specific in the non-quantitative ERα-RIME pull-down analysis were also identified by qPLEX-RIME with significant enrichment over the IgG controls (mean fold-change of 2.5). Notably, the application of qPLEX-RIME achieved overall better peptide coverage for the overlapping ERα-associated proteins compared to the non-quantitative RIME method (Fig. 2d). Additionally, qPLEX-RIME identified 124 more known BioGRID and STRING interactors compared to the non-quantitative RIME

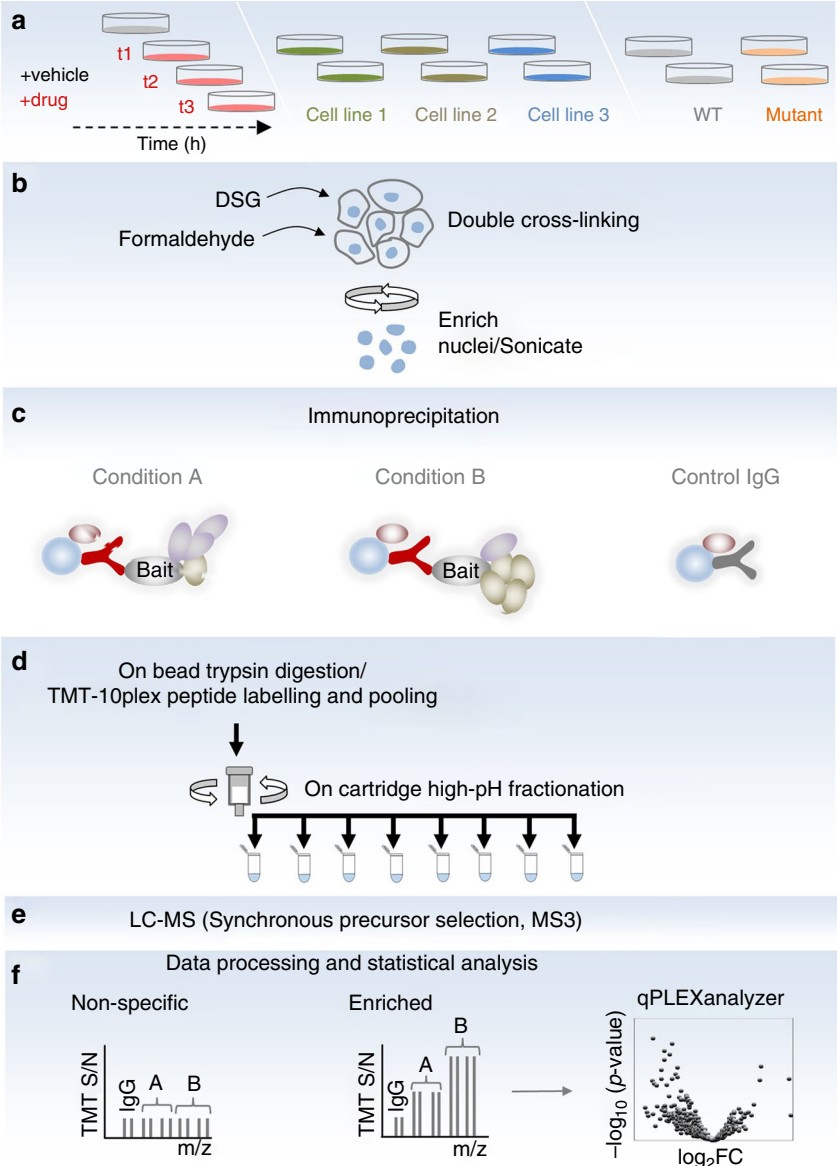

**Fig. 1** The qPLEX-RIME workflow. Proteins in treated and un-treated cell cultures at different time points or in variable genomic backgrounds (e.g. different cell lines or mutated conditions) are double-crosslinked and cell nuclei are isolated and sonicated (**a**, **b**). Target protein complexes are immunoprecipitated and subjected to on-bead trypsin digestion (**c**, **d**). The generated peptides are labelled using different TMT reagents and pooled in a single mixture, which is fractionated using Reversed-Phase cartridges (**d**). Peptide fractions are analysed with the MultiNotch MS3 method (**e**) followed by data processing and statistical analysis using novel analytic suite (qPLEXanalyzer) (**f**)

analysis (175 proteins > 2-fold and adj. *p*-value < 0.01 (Limma moderated *t*-test) in qPLEX-RIME versus 51 proteins in non-quantitative RIME). Importantly, using the qPLEX-RIME we identified a number of novel ERα-associated candidate proteins. We validated the interactions of CBX3 (HP1γ), NIPBL and FOXK1 with ERα, using Proximity Ligation Assay (PLA)[29] (Supplementary Fig. 1a). A GFP negative control was used to moninor for non-specific interactions (Supplementary Fig. 1b). Treatment of the MCF7 cells with the Selective ERα Degrader (SERD) Fulvestrant[30] (Supplementary Fig. 2) disrupted the above interactions demonstrating the specificity of the PLA assay and validating the interactors discovered by qPLEX-RIME (Supplementary Fig. 1a and c).

To test whether our quantitative pipeline can be widely used to study interactors of different bait proteins, we performed qPLEX-RIME experiments on three additional factors following the same experimental design as above. For these and all subsequent

experiments described in this study, we adopted the double crosslinking approach as a comparison between single and double crosslinking for ERα qPLEX-RIME data showed that the latter increases the pull-down efficiency of known and previously validated ERα interactors, including FOXA1, NR2F2 and NCOR2 (Supplementary Fig. 3a and Supplementary Data 3).

Firstly, the qPLEX-RIME method was applied to explore the interactome of CBP (CREB-binding protein) and NCOA3 (SRC-3); two well-characterised co-activators of nuclear receptors[31]. We identified 1437 and 1135 proteins for CBP and NCOA3, respectively, in the two multiplexed sets of bait and IgG pull-downs at peptide FDR < 1% (Supplementary Data 4 and 5). Both bait proteins were highly enriched in the target pull-downs compared to the IgG controls (CBP:$\log_2$fold-change = 3.2 and NCOA3:$\log_2$fold-change = 3.39) with a high number of unique peptides (44 unique peptides for CBP and 36 unique peptides for NCOA3) (Fig. 3a, b). Known interactors of CBP and NCOA3

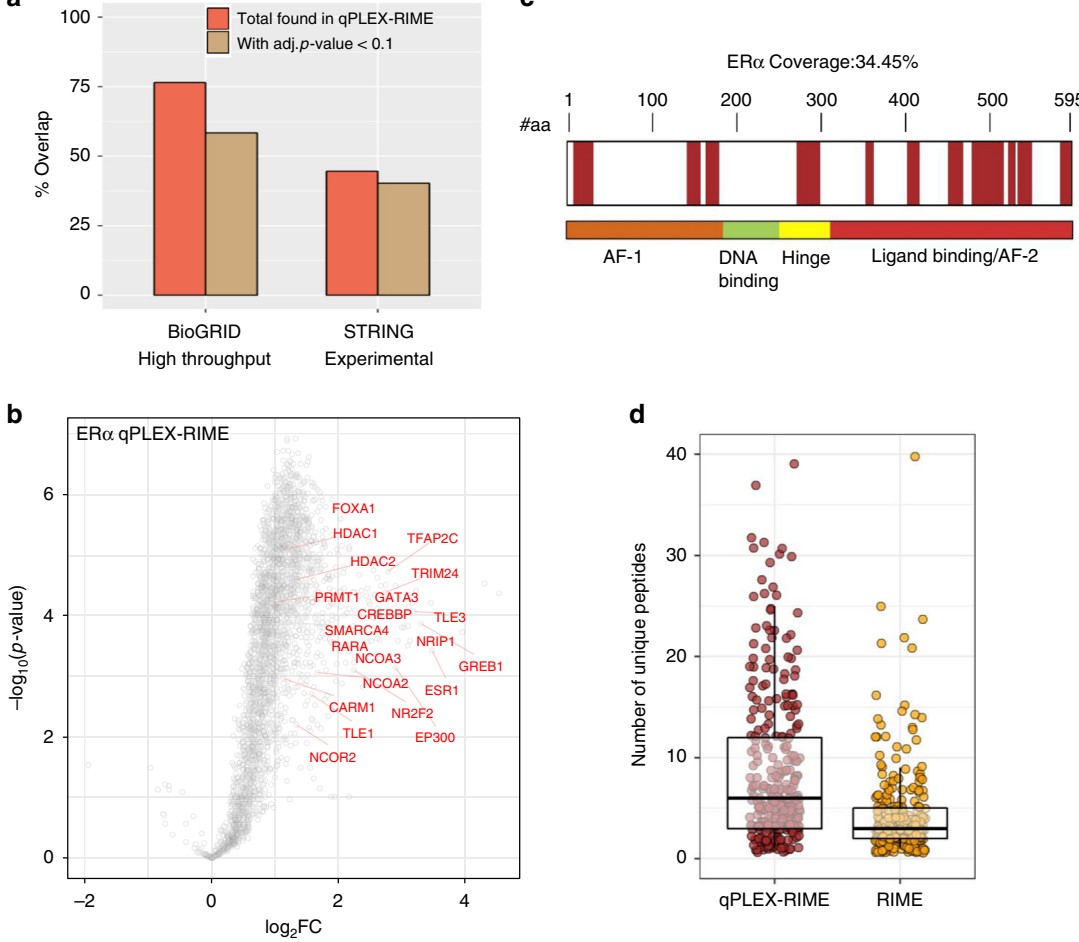

**Fig. 2** Application of qPLEX-RIME for the identification of ERα-specific interactors in breast cancer cell lines. **a** Bar plots illustrating the overlap of the qPLEX-RIME data with known ERα-associated proteins from BioGRID and STRING databases. **b** Volcano plot summarising the quantitative results of the ERα qPLEX-RIME. ERα and several of its known significantly enriched interactors are labelled. **c** Sequence coverage of the ERα protein in the qPLEX-RIME analysis. **d** Boxplots of the number of unique peptides for the overlapping ERα-associated proteins between the non-quantitative RIME and the qPLEX-RIME method. Centre line shows the median, bounds of box correspond to the first and third quartiles and the upper and lower whiskers extend to the largest or the smallest value no further than 1.5 × IQR (inter-quartile range)

were identified including EP300, p160 co-activators, arginine methyltransferases and the ERα complex[31] (Fig. 3a, b). We also identified several members of the SWI/SNF chromatin remodelling complex, such as SMARCA4 (BRG1), SMARCE1 (BAF57), SMARCB1 (BAF47) and SMARCC2 (BAF170). Additionally, in the CBP qPLEX-RIME experiment we captured the association of CBP with the transcription factor JunB[32], as well as with subunits of the mediator complex[33], which are known to associate with enhancer regions as well. Interestingly, in addition to other co-activators, we also found a strong enrichment of co-repressors such NCORs and HDACs in both data sets. This suggests that both co-activators and co-repressors are part of the same complex, which is consistent with previous findings demonstrating extensive co-localisation of co-repressors and co-activators by ChIP-seq[34].

Secondly, we studied the interactome of phospho-RNA polymerase II (POLR2A) using an antibody that recognises the phosphorylated serine-5, which serves as a platform for assembly of factors that regulate transcription initiation, elongation, termination and mRNA processing[35]. We identified 1442 proteins across all multiplexed samples (Supplementary Data 6) and the bait protein was one of the top enriched proteins (log₂fold-change = 4.2), identified with 96 unique peptides

(Fig. 3c). A list of known polymerase II-associated factors were also observed, such as subunits of the SWI/SNF complex, proteins of the mediator complex, initiation and elongation factors[36,37] that are highlighted in Fig. 3c. A comparison of the interactomes of the four bait proteins (ERα, CBP, NCOA3 and POLR2A) showed significant numbers of uniquely identified interactors as well as partial overlap (Supplementary Fig. 3b). To examine whether the overlapping proteins are more likely due to the common underlying biology of the four baits rather than technical bias, we made a venn diagram using a random selection of proteins identified in the four qPLEX-RIME experiments, without considering enrichment relative to the IgG pull-downs (Supplementary Fig. 3c). This analysis showed a smaller number of proteins in the intersection of the four bait proteins indicating small contribution of technical factors to the observed overlap.

Taken together, our data demonstrate a gain in sensitivity using the qPLEX-RIME method that can lead to the identification of interacting proteins with statistical robustness and can be widely used for the characterisation of different interactomes.

**Study of ERα complex dynamics upon OHT treatment.** To investigate the dynamics of the ERα complex assembly upon

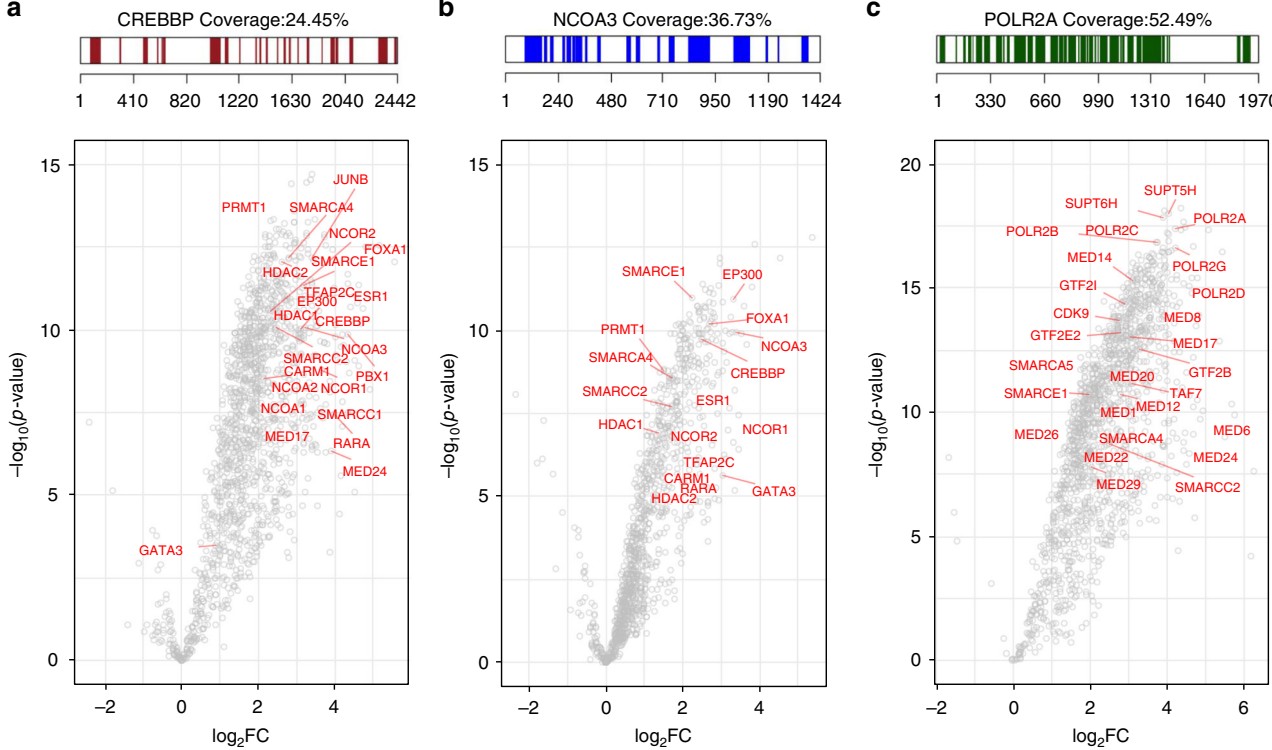

**Fig. 3** Application of qPLEX-RIME in CREBBP, NCOA3 and POLR2A. **a** Peptide sequence coverage of the CREBBP (CBP) protein in the qPLEX-RIME analysis (top panel). The volcano plot summarises the quantitative results of the CBP qPLEX-RIME and several of its known interactors are highlighted in red font (bottom panel). **b** Peptide sequence coverage of the NCOA3 protein in the qPLEX-RIME analysis (top panel). The volcano plot summarises the quantitative results of the NCOA3 qPLEX-RIME and several of its known interactors are highlighted in red font (bottom panel). **c** Peptide sequence coverage of the phospho-polymerase II (POLR2A) protein in the qPLEX-RIME analysis (top panel). The volcano plot summarises the quantitative results of the POLR2A qPLEX-RIME and several of its known interactors are highlighted in red font (bottom panel)

treatment with the Selective Estrogen Receptor Modulator (SERM) 4-hydroxytamoxifen (OHT), we performed three qPLEX-RIME experiments ($3 \times 10$plex) using independently prepared biological replicates. MCF7 cells were crosslinked after treatment with 100 nM OHT for 2 h, 6 h and 24 h or after 24 h of vehicle (ethanol) treatment. Two biological replicates of each condition were included in each experiment, resulting in a total of six replicates per time point. Additionally, MCF7 cells were treated with OHT or ethanol and crosslinked after 24 h treatment in each experiment to be used for control IgG pull-downs, to enable discrimination of non-specific binding.

To confirm that the drug treatment was successful, we performed RNA-seq analysis of six biological replicates using matched OHT treated samples. The mRNA data revealed transcriptional repression of a number of known ERα target genes at 6 h and 24 h, confirming the response to the drug treatment. Specifically, at 24 h treatment the expression of *PGR*, *PDZK1*, *TFF1*, *AREG*, *PKIB*, *SIAH2*, *MYB*, *HEY2*, *FOS*, *GREB1* and *TFF3*[38–43] was significantly inhibited compared to the vehicle treatment (log$_2$Fold-Change $< -0.5$, adj. *p*-value $< 0.05$, Limma moderated *t*-test) (Supplementary Fig. 4a and Supplementary Data 7).

MultiNotch MS3 analysis of the qPLEX-RIME samples quantified 1105 proteins (FDR $< 1\%$) across all three replicate experiments. Of these, 412 proteins were significantly enriched in ERα pull-downs compared to IgG samples (log$_2$Fold-Change $> 1$, adj. *p*-value $< 0.01$, Limma moderated *t*-test) (Supplementary Data 8). Total ERα levels changed upon OHT treatment (Supplementary Fig. 4b), indicating that altered levels of antigen may influence the amount of purified proteins. Our data showed that this resulted in a significant dependency of the quantified

proteins on the amount of ERα pulled down (Supplementary Fig. 5a). To correct for this effect, we applied a linear regression approach[15,44] using the ERα profile as the independent variable and the profile of any other protein as the dependent variable. The advantage of this approach is that proteins with strong dependency on the target protein are subjected to significant correction, whereas proteins with small dependency on the target protein are only slightly corrected. Two such examples, of known ERα interactors before and after correction are shown in Supplementary Fig. 5b. Finally, using the quantification values corrected for the abundance of ERα, we found 249 specific proteins with altered profile in the interactome in at least one time point ($|$log$_2$Fold-Change$| > 0.5$, adj. *p*-value $< 0.05$, Limma moderated *t*-test) allowing for a comprehensive mapping of the dynamic organisation of the ERα complex in response to OHT treatment.

**Dissociation and recruitment of co-factors upon OHT treatment**. We next interrogated the significant changes observed in the ERα interactome at each time point during OHT-mediated growth inhibition. After 2 h treatment with OHT, a significant loss of 12 proteins was observed including known ERα co-activators, such as NCOA3 (AIB1/SRC-3) and CREBBP (CBP) (Fig. 4a). These proteins have been associated with histone acetylation and activation of gene transcription[23,45] and their loss in the ERα interactome upon OHT treatment is consistent with previous studies showing that OHT binding blocks access of co-activators[46]. We also observed a significant loss of the interaction between ERα and NRIP1 (RIP140) protein. NRIP1 can act as a corepressor or as a coactivator[47] with previous evidence

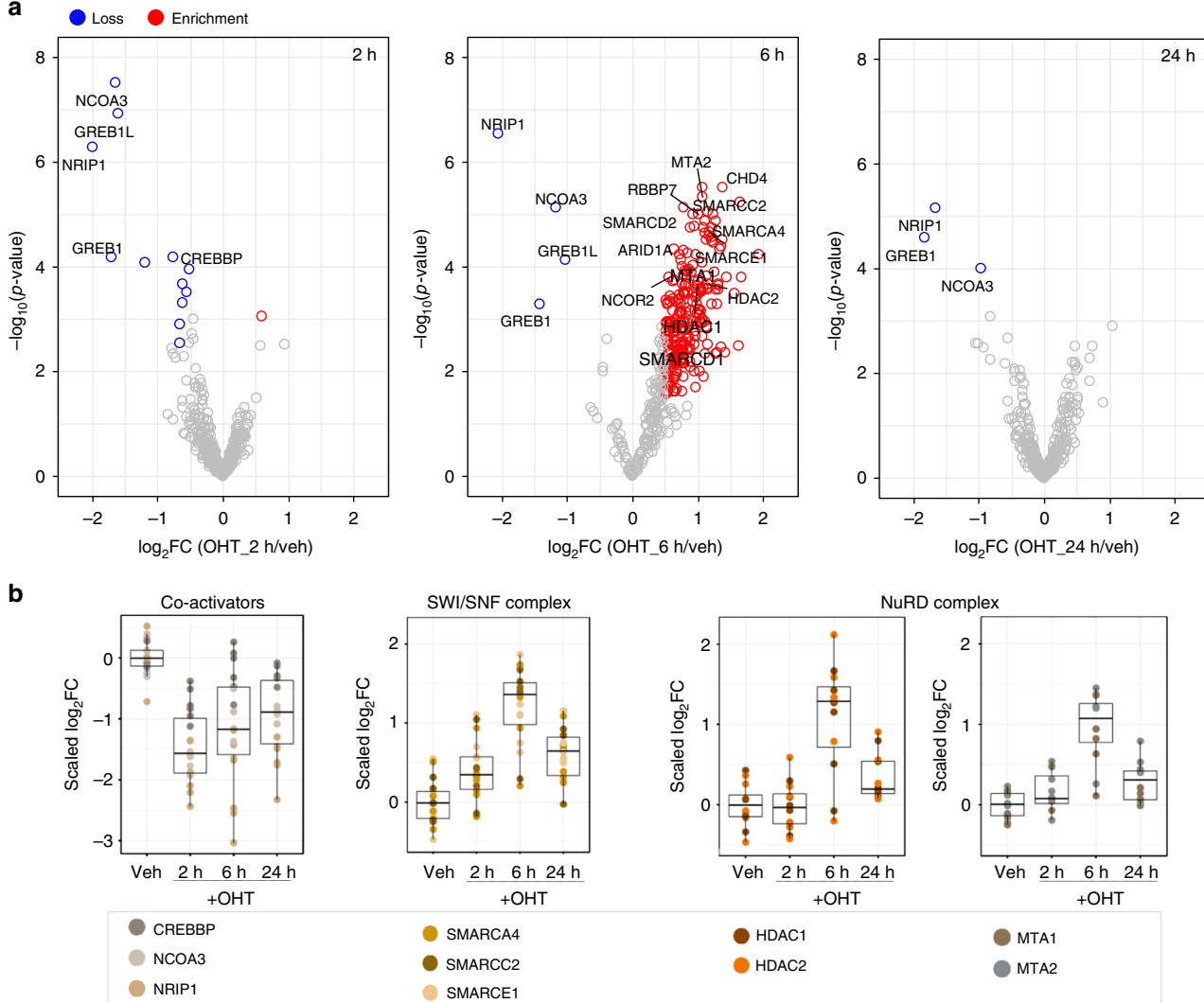

**Fig. 4** Temporal profiling of the ERα interactome following treatment of MCF7 cells with OHT. **a** Volcano plots highlighting enriched or lost proteins in the ERα interactome upon OHT treatment for 2 h, 6 h and 24 h. **b** Boxplots illustrating the loss of ERα co-activator proteins CREBBP, NCOA3 and NRIP1 at 2 h and the enrichment of several subunits of the SWI/SNF and NuRD complexes at 6 h (left to right). Quantitative values are normalised so that the median of the vehicle treated samples is zero (centered around the median of vehicle). Centre line shows the median, bounds of box correspond to the first and third quartiles and the upper and lower whiskers extend to the largest or the smallest value no further than 1.5 × IQR (inter-quartile range)

suggesting that NRIP1 is required for ERα-complex formation and ERα-mediated gene expression[47]. The quantification profile of these key ERα co-activator proteins across all biological replicates is shown in Fig. 4b. Furthermore, OHT treatment resulted in the loss of GREB1 and its paralog gene product GREB1L. Loss of GREB1 upon OHT treatment has been previously described[13], but here we report the loss of both proteins simultaneously.

After 6 h treatment with OHT, 237 specific interactors showed significant enrichment (log$_2$Fold-Change > 0.5, adj. $p$-value < 0.05, Limma moderated $t$-test) compared to the vehicle treatment, whilst NCOA3, NRIP1, GREB1 and GREB1L remained at decreased levels in the interactome (log$_2$Fold-Change < -0.5, adj. $p$-value < 0.05, Limma moderated $t$-test) (Fig. 4a). Notably, there was an enrichment in the recruitment of several components of the NuRD (Nucleosome Remodelling and Deacetylase) complex, e.g. HDAC1/2[24,48] and the signature components MTA1/2[24,48], as well as an enrichment of the co-repressor NCOR2 (SMRT)[25,49] (Fig. 4b). Consistently, NURD complex and NCOR2 has been previously shown by ChIP to be

recruited to promoter regions of ERα target genes following OHT treatment[23,24]. Additionally, we found enriched subunits of the ATP-dependent chromatin remodelling complex SWI/SNF, which is known to regulate both gene activation and gene repression[50,51]. Detected components included SMARCC2 (BAF170), SMARCE1 (BAF57) and SMARCA4 (BRG1) (Fig. 4b). SMARCA4 protein, which was previously shown to be required for repression of ER-mediated transcription[50], was one of the top enriched SWI/SNF proteins. The loss of NCOA3 and CBP at 2 h and the enrichment of SMARCC2 (BAF170) and HDAC1 at 6 h was validated with PLA assays (Supplementary Fig. 6a and b).

At 24 h we observed an almost complete restoration of the ERα complex, with the exception of the NCOA3, NRIP1 and GREB1 proteins, which were still decreased (log$_2$Fold-Change < −0.5, adj. $p$-value < 0.05, Limma moderated $t$-test) (Fig. 4a). Taken together, our results indicate that the inhibitory effect of OHT peaks at 6 h, where ATP-dependent remodelling and corepressor complexes may coordinate to create a transcriptionally inactive chromatin environment.

**Identification of net changes in the ERα complex**. Our data suggest that treatment of MCF7 cells with OHT triggers significant changes in the composition of ERα interactome. To assess whether the changes identified by the qPLEX-RIME analysis are specific changes in interactions or result from changes in total protein levels, we performed timecourse whole proteome quantification in matched samples under the same conditions (vehicle, 2 h, 6 h and 24 h, four biological replicates each) (Supplementary Data 9). We confirmed the OHT up-regulation of ERα protein levels ($\log_2$Fold-Change: 2 h 0.21, 6 h 0.5 and 24 h 1), which was not due to an increase in gene transcription. This is consistent with previous reports demonstrating increased ERα stability in the presence of OHT[52]. A comparison between the qPLEX-RIME results and the total proteome data confirmed that the changes detected in the ERα complex upon OHT treatment represent changes in protein recruitment as the respective total protein and mRNA levels remained unchanged (Fig. 5a). GREB1 was the only ERα interactor with decreased mRNA and total protein levels at 24 h treatment. This is consistent with GREB1 being an ERα target gene[13,43] and explains the decreased association between ERα and GREB1 at this late time point.

Downstream k-means clustering of the most variable proteins (adj. p-value < 0.05, Limma moderated t-test) across the three time points in the total proteome, identified clusters of up- and downregulated proteins (Fig. 5b). Gene Set Enrichment Analysis of the clusters, performed in Perseus software[53], displayed an overrepresentation of genes related to estrogen response and tamoxifen resistance (Fig. 5c). Our findings also revealed the downregulation of proteins involved in cell cycle[54] (Supplementary Fig. 7a), in line with the antiproliferative effects of OHT[24]. Overall, significant changes in gene expression were observed already at 6 h coinciding with pronounced changes in the ERα interactome. As expected, the most significant changes in the total proteome were observed at the later time point (24 h). These results confirm that shuffling of ERα-associated proteins is not typically due to global changes in protein levels. The low mRNA-to-protein correlation at 2 h and 6 h and the respective strong correlation at 24 h are shown in Supplementary Fig. 7b and c. We conclude that our qPLEX-RIME data in combination with the total proteome measurements delineate both the local molecular events in the ER interactome and the associated downstream global effects.

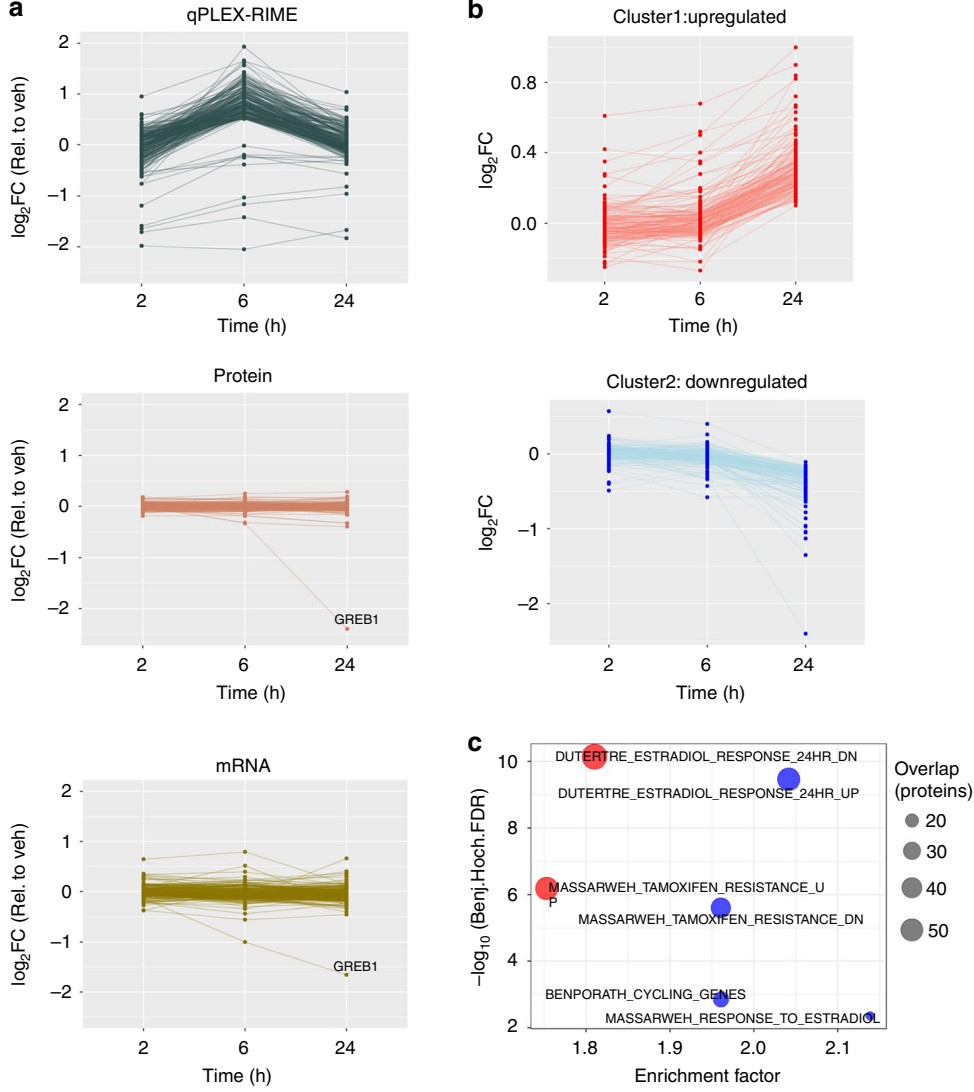

**Fig. 5** Comparison of qPLEX-RIME data with total proteome analysis and RNA-seq data. **a** Line plots of the significantly enriched or lost proteins in the qPLEX-RIME data (top panel), their respective profiles in the total proteome analysis (middle panel) and in the RNA-seq analysis (bottom panel). **b** Line plots representing two k-means clusters of down- and upregulated proteins identified in total proteome analysis (top 10% most variable proteins with at least one significant change, adj. p-value < 0.05). **c** Gene set enrichment analysis for the down- and upregulated protein clusters

**Application of qPLEX-RIME in clinical tumour material**. To test whether qPLEX-RIME can be used to capture chromatin-associated protein–protein interactions in cancer specimens, we conducted an ERα qPLEX-RIME experiment using three independent ER positive human PDX tumours (HCI-003, HCI-005, HCI-006) that have been previously described[55]. Cryosections (30 μm) of each tumour were double-crosslinked and each tumour was split into ERα and matched IgG pull-downs (Fig. 6a). The MultiNotch MS3 analysis identified 2319 proteins (FDR < 1%) across all multiplexed samples with highly reproducible profiles (Supplementary Fig. 8a and Supplementary Data 10). This analysis successfully recovered and quantified ERα

($\log_2$Fold-Change = 1.72, adj. $p$-value = 0.026, Limma moderated $t$-test, unique peptides = 3) using an unbiased mass spectrometry approach in tissue. In addition, many validated and known ERα interactors were discovered from the qPLEX-RIME conducted in PDX material, including CBP[23,56], NCOA2[56], HDAC1[24], GREB1[13], SMARCE1 (BAF57)[51], SMARCA4 (BRG1)[45] and NCOA5 (CIA)[57] (Supplementary Fig. 8b). Sequence analysis of the qPLEX-RIME data showed that 60% of the significant interactors were identified with at least one unique human peptide (i.e. a peptide that does not align to the mouse proteome), indicating that the proteins identified above were primarily from the human cancer cells. Consistently, the tumour samples showed high

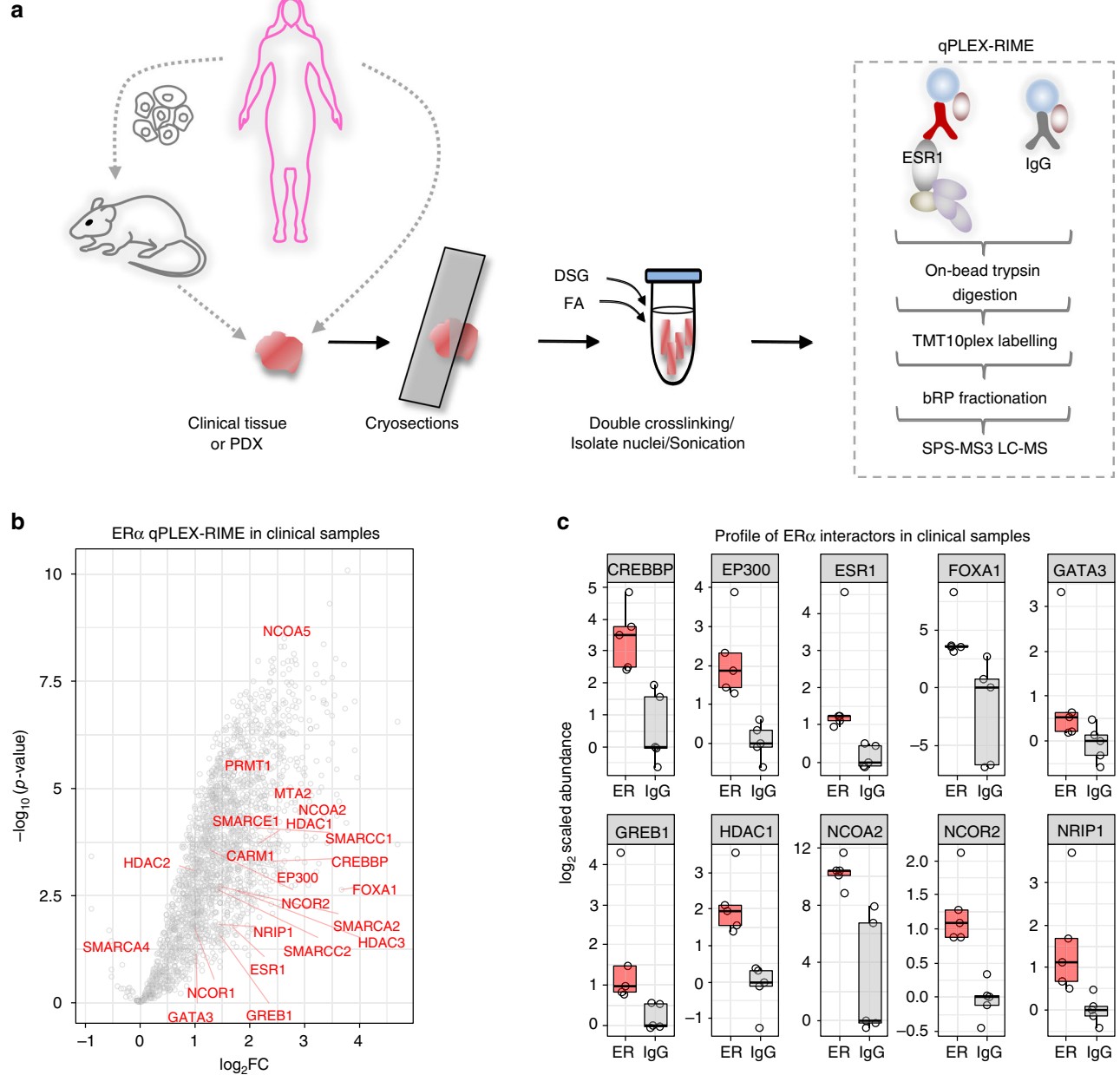

**Fig. 6** Characterisation of ERα interactors from in vivo samples. **a** Schematic representation of the ERα qPLEX-RIME workflow in human xenograft tissues or in human breast cancer tumours. **b** Volcano plot summarising the quantitative results of the ERα interactome in human breast cancer tumours. Several well-known ERα interactors are labelled. **c** Boxplots illustrating the enrichment of selected known ERα interactors in the ERα samples compared to IgG controls in human breast cancer tissues. The $\log_2$ values are normalised so that the median of IgGs is zero. Centre line shows the median, bounds of box correspond to the first and third quartiles and the upper and lower whiskers extend to the largest or the smallest value no further than 1.5 × IQR (inter-quartile range)

cellularity and positive staining for human ERα exclusively in the cancer cells and not in the stroma (Supplementary Fig. 8c).

Prompted by the successful application of qPLEX-RIME in PDX tumours we sought to test the sensitivity of our method in human cancer clinical tissues, collected from surgery. To this end, we performed an ERα qPLEX-RIME experiment in five independent human breast cancer tumours (ERα positive, PR positive, Her2 negative and Grade2/Grade3). Approximately 60 sections (30 μm) were obtained per sample, which were double-crosslinked and split for ERα and matched IgG pull-downs (Fig. 6a). The analysis successfully recovered ERα with excellent coverage (17 unique peptides), as well as 2191 proteins (FDR < 1%) that were quantified in all samples combined (Supplementary Data 11). These included well-described ERα interactors such as FOXA1, GATA3, GREB1, EP300, CBP, HDACs, NCORs and NCOA2 and subunits of the SWI/SNF complex (Fig. 6b). The enrichment of several ERα interactors in the bait samples compared to IgG control samples is illustrated in Fig. 6c.

Our data highlights the method's sensitivity and ability to identify endogenous protein networks from heterogeneous human tumour samples. Importantly, we report the identification of interactors from human tumour tissue material in an unbiased manner.

## Discussion

Here, we describe qPLEX-RIME, a proteomic method which enables comprehensive mapping of endogenous protein interactomes with high sensitivity and statistical robustness. The qPLEX-RIME approach integrates the well-established RIME immunoprecipitation method with advanced high-resolution quantitative multiplexed mass spectrometry analysis. The method can be utilised to discriminate enriched bona fide binding partners from contaminant proteins and to delineate the dynamics of chromatin-associated protein complexes with in-depth protein detection, reproducible quantification and increased sample throughput. The filtering criteria for the prioritisation of the best candidates depend on the type of experiment and the biological question. For bait proteins where very little is known about their interactome, we recommend the use of more stringent specificity criteria in terms of enrichment fold-change, p-value and number of unique peptides in combination with additional filtering based on functional annotations. When the focus is on the dynamic changes of interactomes, the prioritisation of the candidates mostly relies on their robust quantitative profiling across different conditions.

The multiplexed analysis of our pipeline eliminates the need to compare multiple data obtained by individual LC-MS runs, thereby increasing the quantification coverage in very low abundant protein interactors that are stochastically captured between independent replicate runs[58]. The ability to combine the labelled peptides derived from multiple samples increased the sensitivity of the method and enabled the characterisation of the ERα interactome in clinical tumours. Whilst interactors have previously been detected from clinical material, this required targeted mass spectrometry-based approaches and has not been done in an unbiased manner before[13]. Additionally, the use of isobaric labelling resolves the difficulties encountered with cell lines that are not compatible with stable-isotope labelled culture media and provides a means for quantitative analysis for clinical samples that are not amenable to in vivo isotopic labelling techniques. Importantly, our isobaric-labelling data demonstrated high reproducibility with previously published SILAC data[13] (Supplementary Fig. 7d), confirming the accurate quantification obtained by the MultiNotch MS3 level mass spectrometry analysis.

Here, we focused on ERα, the major driving transcription factor in luminal breast cancer[56], which can be targeted by tamoxifen, a drug used for the treatment of ER$^+$ breast cancer[59]. Although many ERα interactors involved in ER-mediated gene expression have been discovered[23,45] our knowledge about their relevance at the tissue level and the impact of tamoxifen on their global association with ERα remains limited. The quantitative data obtained by the qPLEX-RIME experiments has provided us with a list of ERα-associated proteins with significant enrichment over the IgG samples. These include transient, indirect or weak interactions, as it is known that ERα associates with a number of different co-activators rapidly in a cyclic fashion[23,45]. As such the final readout of the crosslinking-based qPLEX-RIME method represents the sum of these interactions. Among these, we validated the interactions between ERα and three proteins; namely CBX3, NIPBL and FOXK1. CBX3 protein is a member of the HP1 protein family, a group of proteins that have been implicated in gene regulation, DNA replication and nuclear architecture[60], whereas NIPBL is a core subunit of the highly conserved protein complex cohesin that has an important role in chromatin structure, gene expression, and DNA repair[61]. The transcription factor FOXK1 belongs to the forkhead family and has an important role in tumorogenesis[62,63]. These findings demonstrate that the gain in sensitivity obtained by qPLEX-RIME can reveal novel ERα interactors. Collectively, we identified a compendium of 253 proteins with consistent presence in all MCF7 data sets (Fig. 7). Importantly, our data show that the vast majority of these ERα-associated proteins (83%) can now be studied either in PDX or in human clinical tissues validating the relevance of these factors in vivo. Additionally, our qPLEX-RIME data on three additional factors, the CREBBP, NCOA3 and the phosphorylated form of POLR2A, highlight the wide applicability of our pipeline.

The application of qPLEX-RIME targeting ERα at multiple time points after OHT treatment, revealed a dynamic change in ERα co-regulators following drug treatment recapitulating and expanding the existing knowledge of OHT mechanism. After 2 h OHT treatment, we observed a loss of important transcriptional co-activators, such as NCOA3 and CBP, whereas at 6 h we observed enrichment on the recruitment of two well-conserved chromatin remodelling complexes, namely the NuRD and the SWI/SNF complex. This coincided with the enrichment of the basal corepressor NCOR2, which assists in the recruitment of HDAC proteins[64]. At the latest time point of 24 h, we observed a restoration of the ERα complex, which may be linked to the half-life of OHT. The exceptions were NRIP1, GREB1 and NCOA3. Interestingly, NCOA3 is amplified in breast cancer[22] and its expression levels have been associated with the effectiveness of tamoxifen treatment[65]. Further, ChIP-seq analysis has revealed that a number of binding sites of NCOA3 are associated with genes with a predictive value for breast cancer patient outcome[56], supporting an important role of this co-regulator in tamoxifen response.

Our timecourse data indicate a switch between activation and repression of transcription in response to OHT treatment. This transition engages a two-step process with the immediate loss of co-activators, followed by the recruitment of co-repressors and ATP-chromatin remodelling complexes that may act cooperatively or in a sequential manner to accomplish transcriptional repression. The integration of qPLEX-RIME data with global protein and mRNA analysis provides a comprehensive view of the activity of a transcription-associated complex over time. A proposed model of OHT mechanism is depicted in Supplementary Fig. 9.

The qPLEX-RIME method can be used to monitor any dynamic changes of interest and importantly can be applied to clinical samples to study tumour evolution, treatment response or

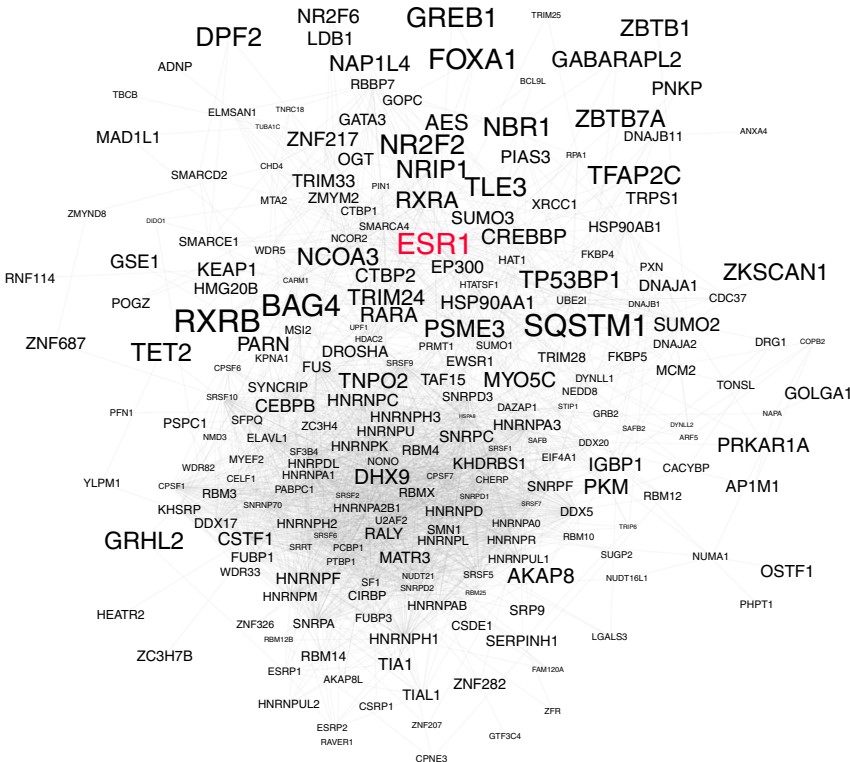

**Fig. 7** Most frequently enriched ERα interactors. STRING network of 253 ERα interactors identified consistently across all the qPLEX-RIME analyses performed in MCF7 cells. The font size increases proportionally to the average fold-change enrichment of these proteins across all the ERα samples compare to IgG controls

numerous other biological and clinical questions. It provides a robust tool for the quantitative analysis of complexes that can be applied to generate comprehensive endogenous protein–protein interaction maps.

## Methods

**Cell lines and cell treatments.** ERα-expressing MCF7 breast cancer cells were cultured in Dulbecco's Modified Eagle Medium DMEM (Gibco, Thermo Scientific, Leicestershire, UK, ref. 41965-239). Media was supplemented with 10% foetal bovine serum (FBS), 50 U/ml penicillin, 50 μg/ml streptomycin and 2 mM L-glutamine. MCF7 cells were obtained from ATCC and they were tested for mycoplasma contamination. Also, the MCF7 cells were genotyped by short-tandem repeat genetic profiling using the PowerPlex_16HS_Cell Line panel and analysed using Applied Biosystems Gene Mapper ID v3.2.1 software by the external provider Genetica DNA Laboratories (LabCorp Specialty Testing Group). For the cell treatments, 4-Hydroxytamoxifen (Sigma-Aldrich, #HG278) or Fulvestrant (Selleckchem, #S1191) were used at final concentration 100 nM.

**Whole cell lysate preparation and western blot analysis.** Cell pellets were reconstituted in 100 μl RIPA buffer (Thermo Scientific, #89901) that was supplemented with protease inhibitors (Roche). 25 μg protein from each sample was loaded on the gel (Invitrogen 4–12%) and the Precision Plus, Protein^TM dual colour Standards Protein molecular weight marker (Bio-Rad, #161-0974) was used for the determination of protein sizes. The proteins were transferred onto a nitrocellulose membrane using the iBlot® 2 Dry Blotting System (Invitrogen) followed by one hour blocking using Odyssey® Blocking Buffer (Li-Cor, 927-40000). The membrane was immunoblotted with ERα antibody (Novocastra #6045332, 1:100) and beta-actin (Cell signalling #4970, 1:1000). Detection of the ER was achieved using the IRDye® 800 CW Goat anti-Mouse (926-32210, Li-Cor Biosciences) diluted to 1:5000, while the loading control was detected using the IRDye 680LT Goat anti-Rabbit (926-68071, Li-Cor Biosciences) diluted to 1:15000. All antibodies were diluted in Odyssey Buffer contained 0.1% Tween. Supplementary Fig. 10 shows the uncropped scan of the blot.

**RNA-seq analysis.** Cells were washed twice with cold Phosphate buffered saline (PBS) and harvested using 350 μl of lysis buffer (RLT). Total RNA was extracted using the RNeasy® kit (Qiagen, #74106) according to the manufacturer's instructions. The extracted RNA was quantified using a NanoDrop® ND-1000

Spectrophotometer (Thermo Scientific). For the library preparation, the Illumina TruSeq Stranded mRNA Library Prep Kit High Throughput was used according to the manufacturer's instructions and two lanes of 50 bp single-end reads were run on HiSeq 4000. Reads were aligned to the human genome version GRCh37.75 using TopHat v2.1.0[66]. Read counts were obtained using feature Counts function in Subread v1.5.2[67] and read counts were normalised and tested for differential gene expression using the DESeq2 workflow[68]. Multiple testing correction was applied using the Benjamini–Hochberg method.

**RIME analysis.** MCF7 cells ($2 \times 10^6$) were grown in complete media. The media was replaced with PBS containing 1% FA (Thermo #28908) and crosslinked for 10 min. For the double crosslinking cells were incubated in PBS containing 2 mM DSG (disuccinimidyl glutarate- Santa Cruz Biotechnology, #sc-285455A) for 20 min followed by incubation in 1% FA for 10 min. Crosslinking was quenched by adding glycine to a final concentration of 0.1 M. For the performance of RIME experiments, 50 μl of Dynabeads® Protein A (Invitrogen) and 5 μg of specific antibody were used for each sample. The antibodies used were: Rabbit polyclonal ERα antibody (Santa Cruz, sc-543), rabbit polyclonal SRC3 antibody (Bethyl laboratories, A300-347A), rabbit polyclonal CBP antibody (Diagenode, C15410224), rabbit polyclonal RNA polymerase II (phospho S5) antibody (Abcam, ab5131) and rabbit IgG antibody (Santa Cruz, sc-2027 or Abcam, ab171870). For nuclear extraction the cell pellet was resuspended in LB1 buffer (50 mM HEPES-KOH (pH 7.5), 140 mM NaCl, 1 mM EDTA, 10% glycerol, 0.5% NP-40 and 0.25% Triton X-100) followed by rotation mixing for 10 min at 4 °C. Then, nuclei were pelleted and resuspended in LB2 buffer (10 mM Tris-HCL (pH 8.0), 200 mM NaCl, 1 mM EDTA and 0.5 mM EGTA) and rotated at 4 °C for 5 min. The samples were resuspended in LB3 buffer (10 mM Tris-HCl (pH 8), 100 mM NaCl, 1 mM EDTA, 0.5 mM EGTA, 0.1% Na-deoxycholate and 0.5% N-lauroylsarcosine). Chromatin was sheared by sonication (Diagenode) to produce DNA fragments of 100–1,000 bp. The bead-bound antibody and chromatin were incubated overnight at 4 °C. The next day the beads were washed 10 times with 1 ml ice-cold RIPA buffer and twice with 500 μl 100 mM AMBIC (ammonium bicarbonate).

**Proximity ligation assay.** Cells were fixed and permeabilised by the addition of ice-cold methanol (−20 °C) for 3 min followed by three washing steps with cold PBS. PLA was carried out according to manufacturer's instructions (Sigma Aldrich, #DUO92007). The following primary antibodies were used for the PLA assay: ERα (Santa Cruz, sc-543 or sc-8002, 1:250) HP1γ (Santa Cruz, sc-365085, 1:400), NIPBL (Santa Cruz, sc-374625, 1:200), FOXK1 (Santa Cruz, sc-373810, 1:200), GFP

(Abcam, ab1218, 1:200), NCOA3 (Bethyl Laboratories, A300-347A, 1:200), CBP (Bethyl Laboratories, A300-363A, 1:200), BAF170 (Santa Cruz, sc-17838, 1:200), HDAC1 (Santa Cruz, sc-81598, 1:200) and the incubation on the coverslips was performed for 1 h at 37 °C. For the single PLA recognition experiment two ERα antibodies (Santa Cruz, sc-543, 1:800 and Invitrogen, MA5-13191, 1:1200) were used in combination. The secondary proximity probes (Sigma Aldrich, Rabbit-PLUS, #DUO92002 and Mouse-MINUS, #DUO92004) were incubated for 1 h at 37 °C. The Leica DFC340FX microscope was used and images were captured with Leica Imaging software. DAPI and PLA fluorescence were captured at high resolution for a total of 8 separate observation fields. Cell numeration and PLA labelling were carried out using Image J software. Cells and red PLA dots were counted using the 'Analyze Particles' function. For each condition at least 200 cells were imaged and analysed. Then, the average value of number of spots per nucleus was calculated. All statistical analyses were carried out by performing Student's *t*-test.

**Immunofluorescence.** Cells were fixed and permeabilised with ice-cold (−20 °C) methanol for 3 min and after fixation cells were blocked in PBS-5 % (w/v) Bovine Serum Albumin (BSA) (Sigma-Aldrich) for 30 min at room temperature. The primary ERα antibody (Santa Cruz, sc-543, 1:250) was diluted in blocking solution (PBS-5 % (w/v) BSA) and incubated on coverslips for 1 h at 37 °C. Afterwards, the coverslips were washed four times in washing buffer (PBS-0.5% Tween). Secondary antibody conjugated to Alexa Fluor 488 (Invitrogen, #A-21206, 1:500) was diluted in blocking solution and incubated on coverslips for 1 h at 37 °C in the dark. Coverslips were then washed again three times in washing buffer and once in PBS.

**PDX propagation and tissue collection.** Viably frozen PDX tumour tissue was propagated in immune-compromised mice. Briefly, 1 mm³ tumour pieces were implanted into the 4th mammary pad of NSG mice. All mice were supplemented with estrogen, using silastic E2 pellets (made in-house) inserted into the dorsal scruff. Twice weekly standard monitoring and tumour measurement was conducted. Once tumours reached appropriate size, ~1000 mm³, mice were sacrificed by cervical dislocation under deep, isoflurane-induced anaesthesia. The tumours were resected, diced and processed by either snap freezing in liquid nitrogen, fixing in 10% neutral buffered formalin solution for subsequent paraffin embedding, embedding in OCT, or viably freezing in FCS supplemented with 5% DMSO.

**Sample preparation of clinical tumour material.** Clinical samples were cryo-sectioned in 30 μm slices using the Leica CM 3050 S cryostat. Tissue sections were fixed in a two-step procedure by adding 2 mM DSG for 25 min. In the same suspension of tissue sections, 1% FA was added for another 20 min without removal of the DSG. Crosslinking was quenched by the addition of glycine to a final concentration of 0.25 M. Samples were centrifuged for 3 min at 2500 g and the supernatant was discarded. Tissue pellets were washed twice with cold PBS and resuspended in 6 ml LB3 buffer (10 mM Tris-HCl (pH 8), 100 mM NaCl, 1 mM EDTA, 0.5 mM EGTA, 0.1% Na-deoxycholate, and 0.5% N-lauroylsarcosine), followed by tip sonication for 12–20 cycles (30 s on, 1 min off) depending on the tumour size. The downstream processing was performed as described above (see RIME method section) and the tissue samples were separated in two parts for the performance of ERα and IgG RIME pull-down assays. Patient and patient-derived tissues used in this work were collected under protocol X13-0133, HREC/13/RPAH/187. HREC approval was obtained through the SLHD (Sydney Local Health District) Ethics Committee ((Royal Prince Alfred Hospital) zone), and site-specific approvals were obtained for all additional sites. Written consent was obtained from all patients prior to collection of tissue and clinical data stored in a de-identified manner, following pre-approved protocols. All animal procedures were carried out in accordance to relevant national and international guidelines and animal protocols approved by the Garvan/St Vincent's Animal Ethics Committee (Animal ethics number 15/10).

**Immunohistochemistry.** FFPE blocks from PDX tumours were sectioned at 4 μm onto Superfrost Plus slides. Immunohistochemistry was carried out using the Leica Bond Autostainer. Sections underwent dewaxing, heat induced antigen retrieval (Leica reagent ER2, 30 mins), and primary and secondary antibody incubations, using ERα antibody (ab108398, Abcam, 1:500) and the EnVision + Rabbit secondary system, respectively. Sections were counterstained with haematoxylin.

**Trypsin digestion and TMT labelling.** A volume of 10 μL trypsin solution (15 ng/μl) (Pierce) in 100 mM AMBIC was added to the beads followed by overnight incubation at 37 °C. A second digestion step was performed the next day for 4 h. After proteolysis the tubes were placed on a magnet and the supernatant solution was collected after acidification by the addition of 2 μl 5% formic acid. The resultant peptides were cleaned with the Ultra-Micro C18 Spin Columns (Harvard Apparatus) according to manufacturer's instructions. The peptide samples were dried with speedvac, reconstituted in 100 μl 0.1 M TEAB (triethylammonium bicarbonate) and labelled using the TMT-10plex reagents (Thermo Fisher) with a randomised design. The peptide mixture was fractionated with Reversed-Phase cartridges at high pH (Pierce #84868). Nine fractions were collected using different elution solutions in the range of 5–50% ACN.

For the total proteome analysis 200 μl of 0.1 M TEAB, 0.1% SDS buffer was added to each cell pellet followed by probe sonication and boiling at 95 °C. Protein concentration was estimated with Bradford assay (BIO-RAD-Quick start) according to manufacturer's instructions. For each sample, 90 μg of total protein were reduced for 1 h at 60 °C by the addition of 2 μL 50 mM tris-2-carboxyethyl phosphine (TCEP, Sigma). Cysteines were blocked for 10 min on the bench with the addition of 1 μL 200 mM methyl methanethiosulfonate (MMTS, Sigma). For peptide generation, trypsin (Pierce #90058) solution was added at ratio protein/trypsin ~30:1 for overnight digestion at 37 °C. The next day peptides were allowed to react with the TMT-10plex reagents (Thermo Scientific) for one hour. The reaction was quenched with 8 μL of 5% hydroxylamine (Thermo Scientific) and the labelled samples were mixed and dried with speedvac concentrator. The TMT mix was reconstituted and fractionated on a Dionex Ultimate 3000 system at high pH using the X-Bridge C18 column (3.5 μm 2.1 × 150 mm, Waters) with 1% gradient. UV signal was recorded at 280 and 215 nm and fractions were collected in a peak dependent manner.

**LC-MS analysis.** Peptide fractions were analysed on a Dionex Ultimate 3000 UHPLC system coupled with the nano-ESI Fusion Lumos (Thermo Scientific). Samples were loaded on the Acclaim PepMap 100, 100 μm × 2 cm C18, 5 μm, 100 Å trapping column with the ulPickUp injection method using the loading pump at 5 μL/min flow rate for 10 min. For the peptide separation the EASY-Spray analytical column 75 μm × 25 cm, C18, 2 μm, 100 Å column was used for multi-step gradient elution. Mobile phase (A) was composed of 2% acetonitrile, 0.1% formic acid and mobile phase (B) was composed of 80% acetonitrile, 0.1% formic acid. The elution method at flow rate 300 nL/min included the following: for 95 min gradient up to 45% (B), for 5 min gradient up to 95% (B), for 8 min isocratic 95% (B), for 2 min down to 5% (B), for 10 min isocratic equilibration 5% (B) at 40 °C. For the clinical sample analysis, a longer gradient separation was used as follows: for 160 min gradient up to 40% (B), for 10 min gradient up to 95% (B), for 8 min isocratic 95% (B), for 2 min down to 5% (B), and for 10 min isocratic equilibration 5% (B). The Lumos was operated in a data-dependent mode for both MS2 and SPS-MS3 methods. The full scans were performed in the Orbitrap in the range of 380–1500 *m/z* at 120 K resolution. The MS2 scans were performed in the ion trap with collision energy 35%. Peptides were isolated in the quadrupole with isolation window 0.7 Th. The 10 most intense fragments were selected for Synchronous Precursor Selection (SPS) HCD-MS3 analysis with MS2 isolation window 2.0 Th. The HCD collision energy was set at 55% and the detection was performed with Orbitrap resolution 60k and in scan range 110–400 *m/z*.

**Data processing and interpretation.** The collected CID tandem mass spectra were processed with the SequestHT search engine on the Proteome Discoverer 2.1 software for peptide and protein identifications. The node for SequestHT included the following parameters: Precursor Mass Tolerance 20 ppm, Fragment Mass Tolerance 0.5 Da, Dynamic Modifications were Oxidation of M (+15.995 Da), Deamidation of N, Q (+0.984 Da) and Static Modifications were TMT6plex at any N-Terminus, K (+229.163 Da) for the quantitative data. Methylthio at C (+45.988) was included for the total proteome data. The Reporter Ion Quantifier node included a TMT 6plex (Thermo Scientific Instruments) Quantification Method, for MS3 scan events, HCD activation type, integration window tolerance 20 ppm and integration method Most Confident Centroid. The consensus workflow included S/N calculation for TMT intensities and the level of confidence for peptide identifications was estimated using the Percolator node with decoy database search. Strict FDR was set at *q*-value < 0.01.

**Bioinformatics Analysis.** We developed an R package (qPLEXanalyzer) to perform downstream data analysis. All analyses were performed using only unique peptides identified with high confidence (peptide FDR < 1%) across all experiments. Peptide-level signal-to-noise (S/N) TMT values were corrected for equal loading across samples using different normalisation approaches based upon the experiment type. For the regression-based correction, unique peptides were aggregated and proteins identified in all the experiments were kept for further analysis. The normalisation on the bait protein level was carried out at protein level using log₂ row-mean scaled values. To filter-out non-specific proteins, a limma-based differential analysis was performed comparing ER and IgG control samples. In the regression analysis, the ERα profile was used as the independent variable (*x*) and the profile of any other protein as the dependent variable (*y*) excluding the IgG controls. The residuals of the $y = ax + b$ linear model represent the protein quantification profiles that are not driven by ERα amount in the pull-down. More details on the normalisation methods used can be found in Supplementary Note 2. The identification of differentially bound proteins was carried out using the limma-based analysis. A multiple testing correction was applied on *p*-value using the Benjamini–Hochberg method to control the FDR.

**Code availability.** The qPLEXanalyzer and qPLEXdata R packages are available at (https://doi.org/10.5281/zenodo.1237825). Both pipelines are described in detail in Supplementary Notes 1 and 2.

**Data availability**. RNA-seq data have been deposited in NCBI's Gene Expression Omnibus[69] and are accessible through GEO Series accession number GSE104872. The mass spectrometry proteomics data have been deposited to the ProteomeXchange Consortium via the PRIDE[70] partner repository with the data set identifier PXD007968. All other data supporting the findings of this study are available from the corresponding authors on reasonable request.

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

## Acknowledgements

The authors would like to thank the Genomics core, the staff in the Proteomics Core especially Valar Nila Roamio Franklin and Carmen Gonzalez Tejedo, the Bioinformatics Core especially Dominique Laurent Couturier and Rory Stark, the Histopathology Core especially Jo Arnold and Bev Wilson, the Biorepository Core and the Instrumentation Core facility at Cancer Research UK. We also would like to thank Igor Chernukhin for uploading the RNA-seq data to GEO. The authors would also like to thank Alana Welm for sharing the PDX models and Elena Provenzano for the evaluation of the IHC staining. We acknowledge the support of the University of Cambridge and Cancer Research UK. We also would like to acknowledge the National Breast Cancer Foundation of Australia (Grant 14001) and the Personalised Medicine for Breast Cancer Biobank, funded by the Sydney Breast Cancer Foundation, National Health and Medical Research Council (NHMRC), TransBCR Centre for research excellence in translational breast cancer research and the McMurtrie family. The Fusion Lumos Orbitrap mass spectrometer was purchased with the support from a Wellcome Trust Multi-user Equipment Grant (Grant # 108467/Z/15/Z). Parts of this work were funded by CRUK core grant (grant numbers C14303/A17197, A19274) to F.M.; Breast Cancer Now Award (grant number 2012NovPR042) to F.M. R.S. is supported by a fellowship from the Novo Nordisk Foundation (NNF 14136). J.S.C. is supported by an ERC Consolidator award and CRUK funding.

## Authors Contributions

E.K.P., R.S., C.S.D.S. and J.S.C., conceptualisation; E.K.P., A.N.H., R.S. and C.S.D.S., methodology; E.K.P., proteomic sample preparation and mass spectrometry; E.K.P., A.N.H. and R.S., RIME experiments; E.K.P., S.O. and R.S., validation assays; E.K.P. and R.S., in vivo experiments; E.K.P., K.K., A.N.H., T.I.R., M.E., R.S. and C.S.R.C., data analysis; K.K. and M.E., development of qPLEXanalyzer; E.K.P., K.K., A.N.H., F.M., M.E., R.S., C.S.D.S. and J.S.C., study design; K.H., K.M.C., A.S. and E.L., collection of human clinical samples and PDX models; R.S., C.S.D.S. and J.S.C., oversaw all experiments; E.K.P., R.S., C.S.D.S. and J.S.C., writing-original draft; All, writing–review, and editing.
