## [Peer Review File · Nature Communications]

Reviewers' Comments:

Reviewer #1:

Remarks to the Author:

The manuscript by Papachristou et al. describes a workflow they term qPLEX-RIME on the example of the ESR1 receptor. They measure the interactome in MCF-7 cells and follow the time course of the interactor recruitment at three time points after treatment with OHT and tamoxifen. They generate paired transcriptome and proteome datasets to assess global changes upon treatment. Finally, they apply their technique to a patient-derived xenograft model.

Already their initial publication of RIME is reminiscent of ChIP-MS approaches. Here, they make use of the newest generation of mass spectrometer and combine it with a commercially available labeling reagent (Thermo). I highly appreciate the development of the analysis script to their qPLEX-RIME approach as a help for other laboratories and would be happy to see these community-oriented solutions more often. Unfortunately, it is only readily usable with a commercially available software and making the modifications for other output files will require some level of bioinformatics expertise. Here, better solutions, like Perseus are on the market (DOI: 10.1038/nmeth.3901).

Overall, the innovative level of this workflow development is low as it has previously been shown (DOI: 10.1016/j.cels.2016.08.009). I agree that this was not for their RIME approach specifically, but it is completely obvious to anybody that this can be adapted.

The authors report new interactors for ESR1. However, it is very important to understand what this exactly means, because I doubt strongly that 1,116 proteins interact with the ESR directly. Rather they immunopurified likely proteins (and complexes) that reside on the same stretch of chromatin. While mentioning this in the title, a better discussion of this effect would be helpful. In the end, we have an overview of chromatin-associated proteins in the vicinity of ESR1, but not necessarily interactors of ESR1. (See material and methods; line 398: up to 1,000 bp = 4 nucleosomes!). This still results in valid PLA assays as the "interactors" are in proximity, but does not describe what we necessarily might understand as interactors.

In fact, a detailed look at Fig. 2D makes me a little suspicious. BIOGRID and STRING already claim 1097 interactors of ESR1 and only 172 (140+32) are recovered by qPLEX-RIME, but 925 are missed. However, the authors suggest 581 (447+134) new interactors. This discrepancy is never discussed. Especially the BIOGRID and STRING database can be used for a more detailed analysis. For example, BIOGRID separates between low throughput (manual validation) and high throughput (like this experiment). Do the candidates taken from STRING only include physical interactions or also co-expression etc.?

I don't understand why the OHT experiment is not represented in overview plots like the tam experiment (Figure 3A compare Figure 3B). A zoom on certain complexes is fine, but it is hard to judge the overall data. Please provide the OHT experiment as in Fig. 3A.

I also have some issues with the data consistency of the manuscript. The authors want to demonstrate the improvements in their workflow, but samples within a single experiment already

failed and are omitted from the analysis. DSG.FA.rep01 is missing in Suppl. Data Set 5. Notably, Supplementary Material, Fig 2 shows that it failed. While this does not compromise the data, it is in my opinion not suited for a reference data set. There is also one replicate of the IgG controls omitted in the analysis of the xenograft model (Figure 5B, Suppl. Data 9) comparing the 3 xenografts to only two IgG replicates. Further, I don't want to overexaggerate the importance of p-values in biological research, but I don't understand why they differ between down- and upregulated proteins of the analysis of a qPLEX-RIME (lines 175 compare lines 187 or Figure 3A at different time points). On a smaller note, while the main text is about TMT10plex, the data processing is done for TMT 6plex (line 517).

There is also a difference between a clinical sample and a PDX. The PDX has been grown to sufficient size in the mouse and it is not clear it represents exactly the clinical state of the tumor. If the method has such a high sensitivity, the authors should try the clinical sample directly. This would indeed be a very encouraging step forward. Currently it is an overstatement as "endogenous protein interactomes from tumour material" have been done – what else are cancer cell lines if not tumour material originally?

Minor comments:

Figure 2A, 3A, 5C it should be "adj. p-value"

Figure 1 step vi, non-specific plot: I would use lower bars for IgG, A and B as they are not enriched compared to the specific plot, but reduced

Figure 3A: Mark ESR1 in the plots. This allows the reader to validate the normalization procedure.

S1A/B: Scale bars missing

S2A: Scale bar missing

S6A: Please use μ instead of u

Reviewer #2:

Remarks to the Author:

This paper by Papachristou et al describes a modified RIME technique (termed qPLEX-RIME) using ER as the bait protein in ER+ breast cancer cells and PDX specimens. Carroll and colleagues first described the original RIME technique – a modified form of IP-mass spectrometry – in relation to the ER interactome in breast cancer cell lines. The modified method described here termed qPLEX-RIME circumvents some of the limitations with the original RIME including silac labelling, inability to multiplex, and a lack of quantitative data. qPLEX-RIME, which uses a modified fixation and uses TMT-

10plex isobaric tags to “barcode” the samples such that multiple samples can be run simultaneously on a tandem mass spectrometer. As proof of principle, the authors systematically apply qPLEX-RIME to ER+ MCF7 cells and compare it side-by-side with the original RIME, compare the qPLEX data to known ER interactors from two databases, measure tamoxifen-induced changes in the ER interactome, compare the method to the total proteome and to RNAseq data, and demonstrate in vivo utility using three ER+ breast cancer PDX samples. The method, while validating previously identified ER interactors, also identified several novel ER associated proteins – these were confirmed using PLA as a secondary method. Numerous other quality control experiments and data analysis are presented. In sum, this is a well-written article describing in detail a new adapted proteomics method for interrogating ER interactors in breast cancer cells, and could have clinical utility in predicting/directing therapy response. Furthermore, the method and informatic pipeline is broadly applicable to other proteins and fields.

Minor: out of curiosity the authors mention several times the assay is unbiased, however variables such as antibody specificity may affect the interactome (in fact the authors recognized an issue with receptor downregulation and compensate for this). While this would not be applicable for comparing replicates and different conditions using a single antibody, it could affect concordance among users, particularly since the sc antibody is no longer available. It might be worth commenting on potential for reproducibility with and utility of different antibodies. Likewise, 30micron cryosections were used from PDX tumor material – as clinical samples may be quite limiting the authors may want to comment on prospective sample size (mg?) needed for analysis and if flash frozen would be sufficient.

Reviewer #3:

Remarks to the Author:

In their manuscript Papachristou et al. describe the development and application of the qPLEX-RIME workflow, which is essentially an improved version of the RIME method that was published before by the authors. The improvement in qPLEX-RIME relates to the introduction of TMT labeling and quantitative analysis for crosslinked endogenous protein complexes.

The qPLEX-RIME approach was used by the authors for characterization of the ERalpha interactome in MCF7 breast cancer cells. Some of the novel interactors were subsequently confirmed by PLA on the endogenous level. The approach was then used to characterize interactome dynamics upon tamoxifen treatment. Correction for bait expression levels and a global proteome/mRNA analysis were introduced to enhance the analysis. These corrections nicely underscore the quality of the work and the careful analysis that was performed.

The application of quantitative proteomics for AP-MS approaches has been around for some time and the benefit for crosslinked complex pull downs is not really unexpected. As already mentioned, the quality of the work is very high, with detailed characterization of the ERalpha interactome. The study is highly timely and corresponds well to the evolution of the protein interactomics field, i.e. interactome dynamics for endogenous proteins. The addition of an analysis pipeline as an R package is a very nice addition ensuring broad applicability of the approach. The introduction of interactome profiling in PDX models is novel and can become a very powerful application of the method.

There are some remarks on the current version of the manuscript. A major concern is the difficulty for readers to assess how broadly applicable the method is. The manuscript shows a highly detailed analysis of the ERalpha interactome in different conditions. While this is clearly of interest to researchers in this field, it remains difficult to estimate the efforts needed to set this up for other proteins. It is unclear at the moment how many proteins can be targeted using this approach. The double crosslinking approach seems powerful but can clearly also interfere with antibody binding by blocking epitopes. It would be good if the authors can show at least two other complexes using this approach. This would also lead to a better understanding of the background with this approach.

The pull down experiments lead to the identification and quantification of almost 3000 proteins. 1116 proteins were shown to be enriched for the ERalpha samples. While many proteins can be expected in classical pull down experiments, especially with high-end MS instruments, the amount of proteins found here is extremely high. Assuming that about 10000 proteins are expressed in a cell line, this would imply that 30% of the proteins is purified in these conditions. Some shotgun proteomics profiling papers report on similar amounts of quantified proteins. Can the authors comment on this? Is this expected from a double crosslinking strategy or is this purely related to sensitivity of the Fusion Lumos instrument. Why was a cut-off of $\log_2\text{FoldChange} > 1$ used? Is there a rationale for using this or is this chosen arbitrarily? The enrichment of more than 1000 proteins in the PDX samples also raises the question on the specificity. It seems evident that there is overlap with known interaction partners. The authors state that 55% of the reproducible interaction partners from MCF7 are found in the PDX dataset. Does this mean that about 130 proteins from the MCF7 dataset are found in the list of 1094 PDX proteins? Are the other candidate proteins unique for the PDX models?

The changes shown for the interactome upon tamoxifen treatment are interesting. The evolution of the interactome is intriguing and seems to be supported in part by literature. Since the differences between the timepoints are really pronounced (12, 237 and 3 differential proteins), it would be good to confirm some of the changes using another strategy. This would ensure that the observation is not an artifact from the approach. Showing the changes in SWI/SNF or NuRD components, and NRIP1/NCOA3 would be a valuable addition to the manuscript. Maybe PLA can be applied for this.

Minor points:

How was the cut-off determined to define the proteins that were lost or enriched in the tamoxifen time study for the 2hr condition? ($\log_2\text{FoldChange} > 0,5, \text{adj. } p\text{-value} < 0.05$).

Application of qPLEX-RIME in PDX models is clearly innovative. In this set-up it is however impossible to compare the advantages of using a tissue 3D environment vs cell culture conditions. A graft of

MCF7 cells (possibly even treated with tamoxifen) would allow such a comparison and would provide a logical intermediary step between the PDX models and the cell culture conditions.

The authors should also discuss how these labeling approaches relate to DIA-based approaches.

Reviewer #1 (Remarks to the Author):

The manuscript by Papachristou et al. describes a workflow they term qPLEX-RIME on the example of the ESR1 receptor. They measure the interactome in MCF-7 cells and follow the time course of the interactor recruitment at three time points after treatment with OHT and tamoxifen. They generate paired transcriptome and proteome datasets to assess global changes upon treatment. Finally, they apply their technique to a patient-derived xenograft model.

Already their initial publication of RIME is reminiscent of ChIP-MS approaches. Here, they make use of the newest generation of mass spectrometer and combine it with a commercially available labeling reagent (Thermo).

1. I highly appreciate the development of the analysis script to their qPLEX-RIME approach as a help for other laboratories and would be happy to see these community-oriented solutions more often. Unfortunately, it is only readily usable with a commercially available software and making the modifications for other output files will require some level of bioinformatics expertise. Here, better solutions, like Perseus are on the market (DOI: 10.1038/nmeth.3901).

We thank the reviewer for commenting on the potentials of broader applicability of our data analysis pipeline. This prompted us to further improve our analysis package. Specifically we have included additional features in our qPLEXanalyzer tool in order to make it compatible with the output from freely available software such as MaxQuant. The new features are explained in the revised version of the qPLEXanalyzer documentation in Supplementary Data Set 1. We agree with the reviewer that Perseus is one of the best solutions for omics data analysis, which we have also used to perform over-representation analysis for the total protein data (Figure 5C). However, the use of Perseus still requires some expertise by the user to decide which steps (normalization, statistics, visualization etc.) to perform to complete the analysis whereas our package offers pre-defined analysis steps tailored for multiplexed quantitative data (qPLEX-RIME) and supported by generation of visual outputs at each step and detailed tutorial (Supplementary Data Set 1). Additionally our pipeline introduces processing steps such as limma normalization and differential expression, within-group normalization and regression-based correction for bait dependency that are not currently supported by Perseus. We would be happy to see these adopted by the proteomics community and implemented in a future version of Perseus.

2. Overall, the innovative level of this workflow development is low as it has previously been shown (DOI: 10.1016/j.cels.2016.08.009). I agree that this was not for their RIME approach specifically, but it is completely obvious to anybody that this can be adapted.

We thank the reviewer for their comment. We believe that our study is not just a simple variation of the above-mentioned study but represents a very different aspect of protein analysis. There are significant differences between the two studies that should be noted. The cited study is excellent for quantification of ubiquitinated individual peptides similar to routine analysis of other PTMs. Our study quantifies proteins captured as they dynamically interact with a target protein over time, which is arguably more challenging both in terms of methodology and data analysis. Importantly, we show the applicability in clinical tumour material for the first time.

3. The authors report new interactors for ESR1. However, it is very important to understand what this exactly means, because I doubt strongly that 1,116 proteins interact with the ESR directly. Rather they immunopurified likely proteins (and complexes) that reside on the same

stretch of chromatin. While mentioning this in the title, a better discussion of this effect would be helpful. In the end, we have an overview of chromatin-associated proteins in the vicinity of ESR1, but not necessarily interactors of ESR1. (See material and methods; line 398: up to 1,000 bp = 4 nucleosomes!). This still results in valid PLA assays as the “interactors” are in proximity, but does not describe what we necessarily might understand as interactors. In fact, a detailed look at Fig. 2D makes me a little suspicious. BIOGRID and STRING already claim 1097 interactors of ESR1 and only 172 (140+32) are recovered by qPLEX-RIME, but 925 are missed. However, the authors suggest 581 (447+134) new interactors. This discrepancy is never discussed. Especially the BIOGRID and STRING database can be used for a more detailed analysis. For example, BIOGRID separates between low throughput (manual validation) and high throughput (like this experiment). Do the candidates taken from STRING only include physical interactions or also co-expression etc.?

We thank the reviewer for their comment, which allowed us to improve the clarity of our manuscript. It is true that the proteins captured by RIME are not always direct interactors but these can also represent distant or weak interactors which are captured during crosslinking and the final readout is the sum of all these combined. This is more pronounced enrichment for ER α which is known to associate rapidly with a number of different coactivators in a cyclic fashion on chromatin (doi:org/10.1016/S0092-8674(00)00188-4, doi:org/10.1016/S0092-8674(03)00934-6). We now refer to these as “ER α -associated proteins” throughout the manuscript. In addition, we note that with the quantitative pipeline we measure protein enrichment rather than presence-absence (also previously suggested by doi: 10.1074/mcp.M114.041012), that allows us to study extended protein networks by capturing weak and transient interactions. Furthermore, based on the very useful comment of the reviewer about the BIOGRID and STRING we have revised the respective section by applying more specific filters to the ER α -related proteins found in the two resources. Previously we used all types of interactions found in the two databases. We have now used only “high-throughput” BIOGRID interactions as suggested by the reviewer and only “experimental” STRING interactions. First, we found very small overlap between the two known BIOGRID and STRING subsets indicative of the lack of a consensus ER α interactome and second we show that qPLEX-RIME achieves up to 76% and 45% of the known ER α -associated proteins from BioGRID and STRING respectively. The respective revised section now reads: “The qPLEX-RIME raw data processing quantified 2,955 proteins across the multiplexed set of all positive and negative samples at peptide FDR<1% (Supplementary Data 4). To test the efficiency of the method in capturing and quantifying previously described ER α -associated proteins, we compiled a list of known ER α interactors from BioGRID and STRING resources. For BioGRID, we used only a subset of 386 proteins identified by high-throughput assays that are similar to the approach used here and for STRING we used only experimental associations (383 proteins, score>200). Noteworthy, only 37 proteins were common between the two reference subsets. The qPLEX-RIME method identified 295 (76%) and 171 (45%) of the known ER α -associated proteins from BioGRID and STRING respectively of which 225 (58%) and 154 (40%) showed positive enrichment at adj.p-value<0.1 (Fig. 2A).”

4. I don't understand why the OHT experiment is not represented in overview plots like the tam experiment (Figure 3A compare Figure 3B). A zoom on certain complexes is fine, but it is hard to judge the overall data. Please provide the OHT experiment as in Fig. 3A.

We thank the reviewer for noticing the inconsistency in the plot labeling. We apologize for not clearly showing that both figures refer to the same experiment, where we have treated

MCF7 with 4-hydrotamoxifen (OHT) in three different time points. We have changed the label from “tam” to “OHT” for figure 4A to make it consistent with the label on Figure 4B.

5. I also have some issues with the data consistency of the manuscript. The authors want to demonstrate the improvements in their workflow, but samples within a single experiment already failed and are omitted from the analysis. DSG.FA.rep01 is missing in Suppl. Data Set 5. Notably, Supplementary Material, Fig 2 shows that it failed. While this does not compromise the data, it is in my opinion not suited for a reference data set. There is also one replicate of the IgG controls omitted in the analysis of the xenograft model (Figure 5B, Suppl. Data 9) comparing the 3 xenografts to only two IgG replicates.

We thank the reviewer for their comment which helped us improve the presentation of our work. We would like to point out that quality control is very critical in interactome data, to avoid data misinterpretation and it is important that our pipeline facilitates the detection of outliers through the use multiple replicates, statistical analysis and visualization. For instance, the reason we excluded the particular DSG.FA.rep01 sample was that we found a strong enrichment of co-immunoprecipitated proteins, which was higher compared to the rest of the respective replicates. Although this is in line with our conclusion that double cross-linking achieves better recovery of ER α -associated proteins it could lead in some over-estimation of the mean fold-changes and we therefore chose to exclude the particular sample. Nevertheless all other replicates included in the multiplexed set show very good reproducibility despite cell collection at different passages and as correctly mentioned by the reviewer the data is not compromised. We chose this dataset as a reference in the qPLEX-RIME documentation for two main reasons. Firstly, to demonstrate the utility of our pipeline in evaluating data quality and secondly because this was our only dataset in this study which included a two-way comparison; between different ER α pull-downs of untreated cells and between IgG and ER α pull-downs so that ER α -associated and condition-dependent enriched proteins could be identified simultaneously. We believed that this type of reference dataset would be more useful to the potential users of our pipeline. Regarding the PDX experiment, we already encountered some technical challenges during sample preparation for this particular tissue possibly due to the tissue composition. This pre-analytical variation was reflected at the protein measurements, thus we chose to exclude this particular sample. This analysis constitutes a proof-of-concept for the utility of our method in the analysis of protein interactomes in tissue. Given the greater heterogeneity of tissue samples, the analysis of a larger number of tissues is warranted. In the revised manuscript we have included data from carefully selected primary surgical human tumours samples. These showed very good reproducibility across all samples with very small pre-analytical variation.

6. Further, I don't want to overexaggerate the importance of p-values in biological research, but I don't understand why they differ between down- and upregulated proteins of the analysis of a qPLEX-RIME (lines 175 compare lines 187 or Figure 3A at different time points).

We agree with the reviewer that it is quite common in gene or protein differential expression to observe almost equivalent number of up- and down- regulated genes or proteins (for example (Supplementary figure 6B & 6C)). However, in our study the qPLEX-RIME measures the enrichment or loss of proteins (not up- and down- regulation) in the ER α interactome upon 4-hydrotamoxifen (OHT) treatment, which is known to cause loss of co-activators and enrichment of co-repressor complexes. Our findings are in line with the literature and add temporal resolution into the molecular events. In other words, the negative and positive values are context-dependent rather than a qPLEX-RIME method attribute or

bias. Additionally, to exclude the possibility of technical artifacts, our experiments were performed in three independent 10plex batches using different cell passages prepared over several weeks' time. In the revised version we also validated several of these observations (loss of NCOA3 and CBP at 2h and enrichment of BAF170 and HDAC1 at 6h) with PLA (Supplementary figure 5A & 5B). In other types of quantitative comparisons, such as the one between single and double crosslinking, the qPLEX-RIME method identifies quite similar numbers of loss and enriched proteins (Supplementary figure 3A) as in this case there is no perturbation of the ER α interactome. In an experiment where we compare IgG negative versus ER α samples it is expected that the distribution will be shifted to positive values provided that the IgG does not have high unspecific binding. This shift towards positive values has also been shown by others (doi:10.1038/nmeth.2703).

7. On a smaller note, while the main text is about TMT10plex, the data processing is done for TMT 6plex (line 517).

We thank the reviewer for their comment. We want to clarify that the precursor mass is the same between TMT10plex and TMT6plex reagents (doi: 10.1021/ac301572t). This is also shown in the TMT10plex Thermo reagents online user guide. Their difference lies in the isotopologue masses of N and C in the tags observed at the MS2 or MS3 level. As such, the processing software has only the TMT6plex as an option in the static modifications list.

There is also a difference between a clinical sample and a PDX. The PDX has been grown to sufficient size in the mouse and it is not clear it represents exactly the clinical state of the tumor. If the method has such a high sensitivity, the authors should try the clinical sample directly. This would indeed be a very encouraging step forward. Currently it is an overstatement as "endogenous protein interactomes from tumour material" have been done – what else are cancer cell lines if not tumour material originally?

We agree with the reviewer comment and have now reworded the text and importantly, we have added data from qPLEX-RIME in surgical tumour samples. In our revised manuscript we have included the analysis results of a qPLEX-RIME experiment using five ER-positive breast cancer clinical samples with the respective number of IgG controls. Indeed, the initial size of the tissues and the number of sections obtained was significantly lower than the PDX material, which however did not compromise the results. The respective text in the section "Application of qPLEX-RIME in Patient Derived Xenograft (PDX) and human breast cancer tumours" now reads: "*Prompted by the successful application of qPLEX-RIME in PDX tumors we sought to test the sensitivity of our method in human cancer clinical tissues, collected from surgery. To this end, we performed an ER α qPLEX-RIME experiment in five independent human breast cancer tumours (ER α positive, PR positive, Her2 negative, Grade2/Grade3). Approximately 60 sections (30 μ m) were obtained per sample, which were double-crosslinked and split for ER α and matched IgG pull-downs (Fig. 6A). The analysis successfully recovered ER α with excellent coverage (17 unique peptides) as well as 2,191 proteins (FDR<1%) that were quantified in all samples combined (Supplementary Data 13). These included well-described ER α interactors such as FOXA1, GATA3, GREB1, EP300, CBP, HDACs, NCORs, NCOA2 and subunits of the SWI/SNF complex (Fig. 6B). The enrichment of several ER α interactors in the bait samples compare to IgG control samples is illustrated in Fig. 6C*".

Minor comments:

1. Figure 2A, 3A, 5C it should be “adj. p-value”

We thank the reviewer for their comment. It is quite common in literature that the volcano plot visualizes $-\log_{10}$ -transformed p-values versus \log_2 fold-changes and that the significant hits above a specific multiple hypothesis testing correction threshold (adj. p-value or FDR) are highlighted. Although the adj. p-values can be used, we chose to use p-values, which are the direct output of the statistical test applied. The adj. p-values or FDR can be slightly different depending on the multiple hypothesis testing correction method used and the use of corrected p-values can change the scaling of the plot.

2. Figure 1 step vi, non-specific plot: I would use lower bars for IgG, A and B as they are not enriched compared to the specific plot, but reduced.

We have updated the plot according to the reviewer's suggestion. However, we want to clarify that the original graphs depict scaled values (emphasizing the difference between samples for the same peptide/protein) rather than absolute values. In other words, the mean absolute S/N can be very different between the plots but here we have normalized by this mean to “bring” all plots in the same scale.

3. Figure 3A: Mark ESR1 in the plots. This allows the reader to validate the normalization procedure.

We thank the reviewer for their comment. We want to clarify that ESR1 has $-\log_{10}$ p-value and \log_2 fold-change equal to zero as it was used as the independent variable in the regression analysis. It represents the bait protein that should not change between samples. As such, we believe that marking ESR1 in the plots could be confusing to the readers. However, if the reviewer still feels that this would improve clarity, we are happy to add it, but we are concerned that it will imply that ESR1 itself wasn't enriched, which it was, but subsequent normalization we used to adjust for differences in antigen levels.

4. S1A/B: Scale bars missing

S2A: Scale bar missing

We have included the scale bars in the PLA images.

5. S6A: Please use μ instead of u

We have replaced u with μ in the supplementary figure.

Reviewer #2 (Remarks to the Author):

This paper by Papachristou et al describes a modified RIME technique (termed qPLEX-RIME) using ER as the bait protein in ER+ breast cancer cells and PDX specimens. Carroll and colleagues first described the original RIME technique – a modified form of IP-mass spectrometry – in relation to the ER interactome in breast cancer cell lines. The modified method described here termed qPLEX-RIME circumvents some of the limitations with the

original RIME including silac labelling, inability to multiplex, and a lack of quantitative data. qPLEX-RIME, which uses a modified fixation and uses TMT-10plex isobaric tags to “barcode” the samples such that multiple samples can be run simultaneously on a tandem mass spectrometer. As proof of principle, the authors systematically apply qPLEX-RIME to ER+ MCF7 cells and compare it side-by-side with the original RIME, compare the qPLEX data to known ER interactors from two databases, measure tamoxifen-induced changes in the ER interactome, compare the method to the total proteome and to RNAseq data, and demonstrate in vivo utility using three ER+ breast cancer PDX samples. The method, while validating previously identified ER interactors, also identified several novel ER associated proteins – these were confirmed using PLA as a secondary method. Numerous other quality control experiments and data analysis are presented. In sum, this is a well-written article describing in detail a new adapted proteomics method for interrogating ER interactors in breast cancer cells, and could have clinical utility in predicting/directing therapy response. Furthermore, the method and informatic pipeline is broadly applicable to other proteins and fields.

1. Minor: out of curiosity the authors mention several times the assay is unbiased, however variables such as antibody specificity may affect the interactome (in fact the authors recognized an issue with receptor downregulation and compensate for this). While this would not be applicable for comparing replicates and different conditions using a single antibody, it could affect concordance among users, particularly since the sc antibody is no longer available. It might be worth commenting on potential for reproducibility with and utility of different antibodies.

We thank the reviewer for their valuable comment. We want to clarify that our pipeline is “unbiased” in the sense that we are studying all possible interactors without preselection followed by targeted analysis. We fully agree with the reviewer that the antibody is an important parameter for qPLEX-RIME experiments that can affect the quality of the data. It is important that the users should validate their antibodies before the qPLEX-RIME experiment. Our lab has evaluated two alternative ER α antibodies for qPLEX-RIME and ChIP experiments that can be used for the study of ER α interactome (Carroll lab, unpublished data). This data is not shown in this study as a separate manuscript is currently in preparation. However, RIME has proven to be a useful tool for antibody validation.

2. Likewise, 30micron cryosections were used from PDX tumor material – as clinical samples may be quite limiting the authors may want to comment on prospective sample size (mg?) needed for analysis and if flash frozen would be sufficient.

We thank the reviewer for their comment. In the revised manuscript we have included a qPLEX-RIME experiment using five ER-positive breast cancer clinical samples (flash frozen) with the respective number of IgG controls. The initial size of the tissues and the number of sections obtained (60 sections) was significantly lower than the PDX material which however did not compromise the results highlighting the sensitivity of the qPLEX-RIME method.

Reviewer #3 (Remarks to the Author):

In their manuscript Papachristou et al. describe the development and application of the qPLEX-RIME workflow, which is essentially an improved version of the RIME method that was published before by the authors. The improvement in qPLEX-RIME relates to the introduction of TMT labeling and quantitative analysis for crosslinked endogenous protein complexes. The qPLEX-RIME approach was used by the authors for characterization of the ERalpha interactome in MCF7 breast cancer cells. Some of the novel interactors were subsequently confirmed by PLA on the endogenous level. The approach was then used to characterize interactome dynamics upon tamoxifen treatment. Correction for bait expression levels and a global proteome/mRNA analysis were introduced to enhance the analysis. These corrections nicely underscore the quality of the work and the careful analysis that was performed.

The application of quantitative proteomics for AP-MS approaches has been around for some time and the benefit for crosslinked complex pull downs is not really unexpected. As already mentioned, the quality of the work is very high, with detailed characterization of the ERalpha interactome. The study is highly timely and corresponds well to the evolution of the protein interactomics field, i.e. interactome dynamics for endogenous proteins. The addition of an analysis pipeline as an R package is a very nice addition ensuring broad applicability of the approach. The introduction of interactome profiling in PDX models is novel and can become a very powerful application of the method. There are some remarks on the current version of the manuscript.

1. A major concern is the difficulty for readers to assess how broadly applicable the method is. The manuscript shows a highly detailed analysis of the ERalpha interactome in different conditions. While this is clearly of interest to researchers in this field, it remains difficult to estimate the efforts needed to set this up for other proteins. It is unclear at the moment how many proteins can be targeted using this approach. The double crosslinking approach seems powerful but can clearly also interfere with antibody binding by blocking epitopes. It would be good if the authors can show at least two other complexes using this approach. This would also lead to a better understanding of the background with this approach.

*We thank the reviewer for their comment and we agree that showing the applicability of our method in other complexes would provide a better understanding of the method's general utility. As such, we have now included three additional qPLEX-RIME experiments on different protein complexes; namely the NCOA3, the CBP and the phospho-Polymerase II complexes. In all three examples, we successfully recovered all bait proteins with excellent coverage as well as a number of known associated interactors with statistical robustness. The respective section in the manuscript now reads: *Firstly, the qPLEX-RIME method was applied to explore the interactome of CBP (CREB-binding protein) and NCOA3 (SRC-3); two well-characterized co-activators of nuclear receptors. We identified 1,437 and 1,135 proteins for CBP and NCOA3 respectively in the two multiplexed sets of bait and IgG pull-downs at peptide FDR<1% (Supplementary Data 6 and 7). Both bait proteins were highly enriched in the bait pull-downs compared to the IgG controls (CBP:log₂fold-change=3.2 and NCOA3:log₂fold-change=3.39) with a high number of unique peptides (44 unique peptides for CBP and 36 unique peptides for NCOA3) (Fig. 3A and 3B). Known interactors of CBP and NCOA3 were identified including EP300, p160 coactivators, arginine methyltransferases and the Estrogen Receptor alpha complex (Fig. 3A and 3B). We also identified several members of the SWI/SNF chromatin remodelling complex such as SMARCA4 (BRG1), SMARCE1 (BAF57), SMARCB1 (BAF47) and SMARCC2 (BAF170). Additionally, in the CBP**

qPLEX-RIME experiment we captured the association of CBP with the transcription factor JunB, as well as with subunits of the mediator complex, which are known to associate with enhancer regions as well. Interestingly, in addition to other co-activators, we also found a strong enrichment of corepressors such as NCORs and HDACs in both datasets. This suggests that both co-activators and co-repressors are part of the same complex, which is consistent with previous findings demonstrating extensive co-localisation of corepressors and coactivators by ChIP-seq. Secondly, we studied the interactome of phospho-RNA polymerase II (POLR2A) using an antibody that recognizes the phosphorylated serine-5, which serves as a platform for assembly of factors that regulate transcription initiation, elongation, termination and mRNA processing. We identified 1,442 proteins across all multiplexed samples (Supplementary Data 8) and the bait protein was one of the top enriched proteins ($\log_2\text{fold-change}=4.2$), identified with 96 unique peptides (Fig. 3C). A list of known polymerase II-associated factors were also observed, such as subunits of the SWI/SNF complex, proteins of the mediator complex, initiation and elongation factors that are highlighted in Fig. 3C.

2. The pull down experiments lead to the identification and quantification of almost 3000 proteins. 1116 proteins were shown to be enriched for the ERalpha samples. While many proteins can be expected in classical pull down experiments, especially with high-end MS instruments, the amount of proteins found here is extremely high. Assuming that about 10000 proteins are expressed in a cell line, this would imply that 30% of the proteins is purified in these conditions. Some shotgun proteomics profiling papers report on similar amounts of quantified proteins. Can the authors comment on this? Is this expected from a double crosslinking strategy or is this purely related to sensitivity of the Fusion Lumos instrument. Why was a cut-off of $\log_2\text{FoldChange}>1$ used? Is there a rationale for using this or is this chosen arbitrarily?

This is a valid point. First, we would like to note that due to multiplexing the final protein list is actually the combination of identifications from all 10 samples together, including the IgGs, rather than the results of a single pull down experiment. This significantly contributes to the number of identified proteins. Particularly, during the double crosslinking, transient or indirect interactions are captured so the final readout is the sum of all these combined. In other words, not only stable but also weak and distant ER α interactors are captured. Additionally, the multiplexing almost eliminates the missing values and proteins that otherwise would have been excluded as non-reproducibly identified or stochastically missed in single-shot experiments, are now used for quantification and are included in the final list. Particularly, the use of high-end MS platform in combination with peptide fractionation further increases sensitivity and enables the quantification of proteins by a single peptide, which however can be reproducible between conditions. Such hits may include known low abundant or sub-stoichiometric interactors. In the simple ER α qPLEX-RIME experiment (5 ER α vs 5 IgGs) if we filter for at least two peptides with at least 2-fold change and FDR<5% only about 770 proteins remain which is a rather reasonable number. Additionally, previously known interactors appear to have on average 3-4 times more peptides compared to other proteins reflecting the gain in sensitivity obtained by fractionation and high-end mass spectrometry. It is to be noted that our method mainly aims to identify dynamic changes of protein complexes. As we demonstrate in the manuscript by the tamoxifen treatment experiment, this dynamic profiling requires the analysis of at least three multiplexed sets for statistically robust results. In this type of experiments we use only target-enriched proteins identified in all the 10plex sets which significantly reduces the number of proteins down to a few hundred which represent the most reproducible interactors (in our example ~400). Regarding the cut-off, this was decided based on the mean fold-change of known interactors from BIOGRID and STRING which was 2-fold.

3. The enrichment of more than 1000 proteins in the PDX samples also raises the question on the specificity. It seems evident that there is overlap with known interaction partners. The authors state that 55% of the reproducible interaction partners from MCF7 are found in the PDX dataset. Does this mean that about 130 proteins from the MCF7 dataset are found in the list of 1094 PDX proteins? Are the other candidate proteins unique for the PDX models?

We thank the reviewer for their comment. We want to clarify that this smaller subset represents the most frequently identified proteins in the MCF7 experiments and using the qPLEX-RIME workflow we can now also study these in tissue material regardless the presence of other proteins. Considering proteins identified in any of the MCF7 experiments, only 80 proteins with nuclear localization seem to be significantly enriched uniquely in the PDX samples (>2-fold, adj.p-val<0.01). However, if the biological question is the identification of unique proteins in PDX tissue, then the analysis of additional breast cancer lines would be required for more conclusive results. However such comparison is beyond the scope of this study as our main goal is to show that our method can quantitatively characterize interactors also in tissue material.

4. The changes shown for the interactome upon tamoxifen treatment are interesting. The evolution of the interactome is intriguing and seems to be supported in part by literature. Since the differences between the timepoints are really pronounced (12, 237 and 3 differential proteins), it would be good to confirm some of the changes using another strategy. This would ensure that the observation is not an artifact from the approach. Showing the changes in SWI/SNF or NuRD components, and NRIP1/NCOA3 would be a valuable addition to the manuscript. Maybe PLA can be applied for this.

We thank the reviewer for their comment that aimed to improve the quality of the data and the conclusions made. We agree that the validation of the temporal changes using an alternative approach would ensure that the changes observed using qPLEX-RIME accurately reflect the composition of the complex at a given time point. For this purpose we have validated the loss of NCOA3 and CBP at 2h and the enrichment of SMARCC2 (BAF170) and HDAC1 proteins at 6h with PLA assay (Supplementary Fig. 5A and 5B).

Minor points:

1. How was the cut-off determined to define the proteins that were lost or enriched in the tamoxifen time study for the 2hr condition? ($\log_2\text{FoldChange} > 0,5, \text{adj. } p\text{-value} < 0.05$).

The $\log_2\text{FoldChange} > 0.5$ or < -0.5 was decided based on the standard deviation of all measurements for the 412 ER α enriched proteins and this threshold was common for all three time points. The adj. $p\text{-value} < 0.05$ was chosen arbitrarily to provide a good balance between specificity and sensitivity for the differentially enriched/lost proteins. However the vast majority of the proteins shown had an adj. $p\text{-value} < 0.01$.

2. Application of qPLEX-RIME in PDX models is clearly innovative. In this set-up it is

however impossible to compare the advantages of using a tissue 3D environment vs cell culture conditions. A graft of MCF7 cells (possibly even treated with tamoxifen) would allow such a comparison and would provide a logical intermediary step between the PDX models and the cell culture conditions.

We thank the reviewer for their valuable comment. The development of an intermediate model between the PDX and the cell culture conditions would certainly be a rational step to enable the monitoring of protein interactome changes upon drug treatment in matched genomic backgrounds. However, given the paucity of robust ER+ organoid or 3D systems, we did not explore this because it would mean asking questions in different cancers, beyond our laboratory's interests or expertise. We believe that a more important issue is the use of qPLEX-RIME for clinical samples, either immediately from surgery or primary/PDX material treated in vivo or ex vivo. To this end, we have now established qPLEX-RIME from both PDX tumour material and from surgical material, permitting explant treatment experiments or PDX treatment experiments. Our lab is currently exploring these options, instead of 3D based methods and our new data clearly show that interactome mapping experiments can be conducted in this physiologically relevant context.

3. The authors should also discuss how these labeling approaches relate to DIA-based approaches.

Data Independent Acquisition (DIA)-based analysis offers an alternative method of quantifying full and partial proteomes to isotopically labelled methods. The most appealing aspect of DIA and SWATH, a specialised version of DIA, is that the acquisition of all peptide precursor and fragment ion data within the limit of detection of the instrument can in principle be extracted. This has advantages in the longer term if the same target protein is pursued for a very large number of samples. However the increased complexity of these MS/MS spectra is a significant limitation of the technique. As a result DIA methods have struggled to compete with the collection of MS/MS spectra using traditional DDA strategies. Specifically we have opted for the latter because of the following reasons:

- SWATH requires that specialized/bespoke libraries are generated for each analysis. A major limitation when using clinical tumour samples.
- Specialized data processing is required compared to more standard Mascot or Sequest searches.
- Sample pre-fractionation, which is required for more complex samples, is poorly compatible with such approaches as it can introduce more variation.

In summary we have used isobaric labelling as we seek for a method with simple implementation, compatible with fractionation and with high robustness to analytical errors while maintaining sensitivity.

REVIEWERS' COMMENTS:

Reviewer #1 (Remarks to the Author):

The provided additional data (additional complexes and clinical samples) and some further improvement (MaxQuant integration) does strengthen the manuscript.

Some minor issues:

-There is still uM instead of μM in Figure S1A, S1B, S2, S5A, S5B. Also in the legends, e.g. Suppl. Fig 7 legend – please check the complete manuscript

-Figure 2 D misses the y-axis label

Reviewer #3 (Remarks to the Author):

The manuscript by Papachristou et al. has significantly improved by this review round. All of my comments were addressed by the authors. I think the dynamics study is very promising and I am happy to see confirmation of these findings by PLA. I still have one important remark which is related to the first two comments raised during my previous review.

I still feel that the specificity of the approach is very low. I understand that by crosslinking you can capture large complexes and that this would cause long lists of specifically enriched proteins. Indeed, as suggested by reviewer 1, proteins residing on the same stretch of chromatin can be enriched as well. The data presented in the manuscript suggests that a large part of the nuclear proteome is enriched in the samples. This is highlighted by the experiments on the additional proteins where long lists of potential partners are also generated. In addition, there is significant overlap in the different interactomes for these proteins. This is probably not unexpected because of their underlying biology. On the other hand, there is obviously a good chance of finding overlap if over a thousand (mainly nuclear) proteins are found. It is good to see that there are some differences in the protein lists for the different proteins, but it would be good to provide a better view on the specificity of these pull-down experiments. A simple way would be to provide a Venn diagram with the overlapping proteins for the 4 bait proteins (e.g. in supplementary info).

While a differential approach for the same bait (with and without a stimulus, i.e. the OHT dataset) clearly reveals interesting biology, this is less clear from the general application where a specific pull down is compared to the IgG controls. As the authors point out, the bait proteins show clear enrichment in the relevant samples, but in most cases there are many other proteins which appear to be more enriched with better p values. This makes it really difficult to select/prioritize proteins for biological follow-up. Can the authors elaborate on how the partners for PLA confirmation were chosen? Maybe based on this, can the authors also suggest a general strategy on how these data should be used for biology? This also holds true for the PDX and clinical samples where equally long protein lists are generated. Comparison with existing interactome databases confirms enrichment of relevant partners in the data but it fails to support prioritization for follow-up. This suggests that the method is not very well suited for discovery of complexes, but rather for validation (when indeed specific) and complex dynamics. In conclusion, at the moment it is hard to see how the data generated with the direct pull down experiments will be used and it would be best if the authors address this in the manuscript.

Reviewer #1 (Remarks to the Author):

The provided additional data (additional complexes and clinical samples) and some further improvement (MaxQuant integration) does strengthen the manuscript.

Some minor issues:

- There is still uM instead of μ M in Figure S1A, S1B, S2, S5A, S5B. Also in the legends, e.g. Suppl. Fig 7 legend – please check the complete manuscript
- Figure 2 D misses the y-axis label

We thank the reviewer for acknowledging the improvements in our manuscript. We have replaced u with μ in the supplementary figures and in the text and we have added the y-axis label on the Figure 2D.

Reviewer #3 (Remarks to the Author):

The manuscript by Papachristou et al. has significantly improved by this review round. All of my comments were addressed by the authors. I think the dynamics study is very promising and I am happy to see confirmation of these findings by PLA. I still have one important remark which is related to the first two comments raised during my previous review.

I still feel that the specificity of the approach is very low. I understand that by crosslinking you can capture large complexes and that this would cause long lists of specifically enriched proteins. Indeed, as suggested by reviewer 1, proteins residing on the same stretch of chromatin can be enriched as well. The data presented in the manuscript suggests that a large part of the nuclear proteome is enriched in the samples. This is highlighted by the experiments on the additional proteins where long lists of potential partners are also generated. In addition, there is significant overlap in the different interactomes for these proteins. This is probably not unexpected because of their underlying biology. On the other hand, there is obviously a good chance of finding overlap if over a thousand (mainly nuclear) proteins are found. It is good to see that there are some differences in the protein lists for the different proteins, but it would be good to provide a better view on the specificity of these pull-down experiments. A simple way would be to provide a Venn diagram with the overlapping proteins for the 4 bait proteins (e.g. in supplementary info).

While a differential approach for the same bait (with and without a stimulus, i.e. the OHT dataset) clearly reveals interesting biology, this is less clear from the general application where a specific pull down is compared to the IgG controls. As the authors point out, the bait proteins show clear enrichment in the relevant samples, but in most cases there are many other proteins which appear to be more enriched with better p values. This makes it really difficult to select/prioritize proteins for biological follow-up. Can the authors elaborate on how the partners for PLA confirmation were chosen? Maybe based on this, can the authors also suggest a general strategy on how these data should be used for biology? This also holds true for the PDX and clinical samples where equally long protein lists are generated. Comparison with existing interactome databases confirms enrichment of relevant partners in the data but it fails to support prioritization for follow-up. This suggests that the method is not very well suited for discovery of complexes, but rather for validation (when indeed specific) and complex dynamics.

In conclusion, at the moment it is hard to see how the data generated with the direct pull down experiments will be used and it would be best if the authors address this in the manuscript.

We thank the reviewer for acknowledging the improvements in our manuscript and for their additional comments. We would like to note that our method serves as a discovery platform and the filtering criteria are chosen according to the biological question. For example, if nothing is known about the bait protein interactome, one should start by applying more stringent cut-offs in terms of fold-change, p-value, number of unique peptides and total expression levels of the protein in the respective cells (either by acquiring full proteome data or using public databases). The latter is because, lower abundant proteins are more likely to be true interactors when found in a pull down (compared to high abundant proteins). For example, if nothing was known about ESR1 interactome and we applied the following cut-offs: $\log_2FC > 1$, $\text{adj.p-value} < 0.01$, at least two unique peptides and excluding the 30% more abundant proteins based on the full proteome, we have a list of 251 (40 with $\log_2FC > 2$) candidates only in which ESR1 is the most enriched. Another 11 well known interactors (GREB1, TLE3, NRIP1, EP300, TFAP2C, MAD2L1, GATA3, NR2F6, RARA, ESRRRA and LDB1) are among the top 20 hits. This means that even if we didn't have any prior knowledge about ESR1 interactors, it is apparent that we would have a shortlist of excellent candidates to start with. The shortlist of candidates can be further filtered based on the interest of the specific project, functional annotations about the candidates and availability of reagents for follow up experiments. The respective text now reads: *'The filtering criteria for the prioritization of the best candidates depend on the type of experiment and the biological question. For bait proteins where very little is known about their interactome, we recommend the use of more stringent specificity criteria in terms of enrichment fold-change, p-value and number of unique peptides in combination with additional filtering based on functional annotations. When the focus is on the dynamic changes of interactomes, the prioritization of the candidates mostly relies on their robust quantitative profiling across different conditions'*. Furthermore, to address the comment about the specificity of the method in the particular experiments, we have performed a comparison between the interactomes of the 4 bait proteins using the following cut-offs: $\log_2FC > 1$, $\text{adj.p-value} < 0.05$, at least two unique peptides. For simplicity, we didn't filter out high abundance proteins in this comparison. The results of the comparison are shown in Supplementary Figure 3B. From these we can see that although there is significant overlap between the 4 bait pull downs, there are also significant numbers of uniquely enriched proteins with specific known examples. For example in the POLR2A qPLEX-RIME experiment we have identified initiation and elongation factors as well as RNA polymerase II subunits that were not identified in the other qPLEX-RIME experiments. To examine whether the overlapping proteins are due to underlying biology as we would expect and not due to the relatively high numbers of nuclear proteins that overlap by chance, we made a venn diagram using a random selection of proteins identified in the 4 qPLEX-RIME experiments. For this we used equivalent numbers of proteins per bait with the ones used for the original venn diagram comparison (Supplementary Figure 3C). We identified very small overlap in this instance which supports the idea that the overlap between the enriched proteins per bait is because of common underlying biology rather than by chance. The respective text now reads: *'A comparison of the interactomes of the four bait proteins (ER α , CBP, NCOA3, POLR2A) showed significant numbers of uniquely identified interactors as well as partial overlap (Supplementary Fig. 3B). To examine whether the overlapping proteins are more likely due to the common underlying biology of the four baits rather than technical bias, we made a venn diagram using a random selection of proteins identified in the four qPLEX-RIME experiments, without considering enrichment relative to the IgG pull downs (Supplementary Figure 3C). This analysis showed a smaller number of proteins in the intersection of the four bait proteins indicating small contribution of technical factors to the observed overlap'*. As we emphasize in the manuscript, the best utility of our method is to study protein interactome dynamics. In this type of experiments we aim to monitor the recruitment or loss of interactors (mostly known) across conditions in a quantitative fashion. In this instance, stringent specificity filtering is not necessary and the

prioritization of the candidates relies on their quantitative profiling, their known functional annotations and the availability of reagents. Here, we selected the candidate proteins for PLA validation based on these criteria. We required good enrichment of protein not previously capture by simple RIME, we performed literature mining and we looked for availability of good antibodies. We believe these are the minimum steps for candidate selection but of course the selection process is specific for each project.